# ADIFF: Explaining audio difference using natural language

**Soham Deshmukh    Shuo Han    Rita Singh    Bhiksha Raj**
Carnegie Mellon University
{sdeshmuk,shuohan,rsingh,bhikshar}@andrew.cmu.edu

## Abstract

Understanding and explaining differences between audio recordings is crucial for fields like audio forensics, quality assessment, and audio generation. This involves identifying and describing audio events, acoustic scenes, signal characteristics, and their emotional impact on listeners. This paper stands out as the first work to comprehensively study the task of explaining audio differences and then propose benchmark, baselines for the task. First, we present two new datasets for audio difference explanation derived from the AudioCaps and Clotho audio captioning datasets. Using Large Language Models (LLMs), we generate three levels of difference explanations: (1) concise descriptions of audio events and objects, (2) brief sentences about audio events, acoustic scenes, and signal properties, and (3) comprehensive explanations that include semantics and listener emotions. For the baseline, we use prefix tuning where audio embeddings from two audio files are used to prompt a frozen language model. Our empirical analysis and ablation studies reveal that the naive baseline struggles to distinguish perceptually similar sounds and generate detailed tier 3 explanations. To address these limitations, we propose ADIFF, which introduces a cross-projection module, position captioning, and a three-step training process to enhance the model's ability to produce detailed explanations. We evaluate our model using objective metrics and human evaluation and show our model enhancements lead to significant improvements in performance over naive baseline and SoTA Audio-Language Model (ALM) Qwen Audio. Lastly, we conduct multiple ablation studies to study the effects of cross-projection, language model parameters, position captioning, third stage fine-tuning, and present our findings. Our benchmarks, findings, and strong baseline pave the way for nuanced and human-like explanations of audio differences.

## 1 Introduction

In the Boston Marathon bombing of 2014, forensic investigators were faced with a challenge – were the various audio recordings purportedly captured of the event (and put up on social media) by various people recordings of the same event, or were they mistaken or fraudulent uploads actually from different events? This anecdote highlights the complexity of audio analysis. One can have different sounding recordings of the same event, similar sounding recordings of different events, and other combinations. To know the relation, one must be able to describe the recordings in relative or comparative terms. This natural human ability is used in various contexts- in audio forensics to verify recording authenticity (Maher (2009); Kraetzer et al. (2007)), crucial for legal investigations. In production and broadcasting, it aids in Audio Quality Assessment (Campbell et al. (2009); Deshmukh et al. (2024a)) to detect subtle variations. In audio generation (Liu et al. (2023); Kreuk et al.), it helps create realistic synthetic audio. Currently, identifying audio differences requires a human listener with expertise in phonetics (Johnson & Johnson (2004)),

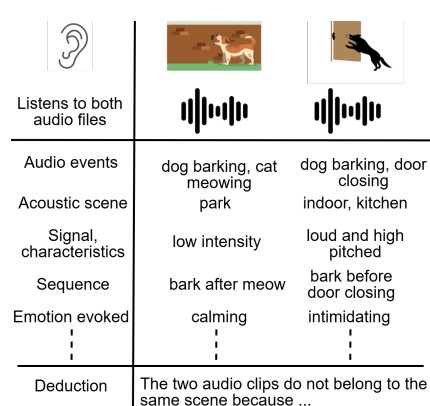

Figure 1: Humans use auditory information to compare scenes and make deductions.

the acoustic-phonetic approach (Stevens (2000)), and spectrogram analysis to examine recordings and identify differences across various parameters.

Recent advancements in audio-text learning has enabled new applications such as text-to-audio retrieval (Koepke et al. (2022); Deshmukh et al. (2023b)), audio captioning (Mei et al. (2022a); Deshmukh et al. (2024b)), and audio question answering (Lipping et al. (2022); Gong et al. (2024)). To excel in these applications, models must possess both fine-grained and coarse-grained understanding of audio, enabling them to describe audio content in human-like natural language. By pretraining models on multiple audio-text tasks, individual audio-text tasks see a benefit in performance improvement (Deshmukh et al. (2023a)). Therefore, the field has shifted towards training audio models capable of performing the aforementioned audio-text tasks using a single model. Examples of such Audio-Language Models (ALM) include SALMONN (Tang et al. (2024b)), GAMA (Ghosh et al. (2024a)), and LTU (Gong et al. (2024; 2023)). However, the current literature has not addressed the task of audio difference explanation. This is due to the challenges in generating coherent paragraph-level descriptions, the lack of cognitive processing and understanding similar to human auditory perception, and the absence of annotated datasets specifically designed for this task.

Explaining the differences between two audio samples also acts as an effective benchmark for assessing a model's comparative reasoning and its ability to integrate audio information with world knowledge. When a model is tasked with distinguishing between two audios, it must employ comparative reasoning to identify both subtle and significant details. This requires the model to understand the fundamental properties of audio signals, such as frequency, amplitude, and temporal patterns, as well as determining pitch, timbre, and loudness. Beyond these signal characteristics, the model must also use world-knowledge to grasp contextual elements, like the genre of music or the type of environment where the audio was recorded. Finally, the model must use comparative and deductive reasoning to interpret and compare these features, identifying nuanced differences and similarities, and draw conclusion. This makes audio difference explanation an effective benchmark for evaluating audio-text models and exploring methods for integrating audio information, world knowledge, and comparative reasoning.

In this paper, our main contributions are:

- Introduce the audio difference explanation task which aims to provide natural language explanation for the differences in two audio files. To build and benchmark models for this task, we create two datasets ACD and CLD, where the difference explanation is generated by LLM using human annotated captions and later verified by human annotators (test set). To mimic human explanations, each dataset contains three tiers of explanation ranging from one-sentence (audio events) to detailed explanations (scene, semantic, listener emotions).

- We propose a naive baseline for this task using prefix-tuning. In this approach, audio embeddings from the two audio files are used to prompt a frozen language model. To address the limitations of this naive baseline, we introduce the ADIFF model, which incorporates separator token, cross-projection layer and undergoes a three-stage training process. These enhancements improve the model's ability to identify detailed differences and distinguish perceptually similar sounds, resulting in a performance improvement over the naive baseline and SoTA ALM (Section 4). The checkpoint will be publicly released[1].

- Under the proposed framework, we conduct several ablation studies to understand the impact of cross-projection, language model scaling, audio-grounding, and fine-tuning. Our findings reveal that (1) cross-projection aids in utilizing text prefixes to store difference attributes, (2) under limited compute and data, smaller language models are easier to ground in audio with proper training and (3) position-based audio captioning with multiple audio inputs enhances the model's performance on similar-sounding acoustic sources. Lastly, we address language model hallucinations by enabling users to detect inaccuracies through comparisons between generated descriptions and predicted audio event probabilities.

## 2    AUDIO DIFFERENCE EXPLANATION

The task of explaining audio differences consists of finding the difference between two audio files and explaining the difference in natural language. Human explanations draw on various sources of information, including acoustic details and human perception, as well as linguistic nuances. From

---

[1]Dataset and pretrained model are available at `https://github.com/soham97/ADIFF`

an acoustic standpoint, explanations can range from broad audio event differences to finer-grained signal variations. They can also be based on objective facts and world knowledge or on human perceptual differences. Linguistically, these explanations can be concise, like a single sentence, or more detailed, like a paragraph. This allows us to segregate audio difference explanations that vary in detail, focus, and style:

**Concise**: This description is concise and straightforward, briefly mentioning the key characteristics of each audio without much elaboration. It highlights the main differences in ambiance and focus between the two audios. We refer to this as the tier-1 explanation.
**Brief**: This answer provides more detail and context. It not only describes the sounds but also includes audio events, sound sources, the nature of the sounds, making it slightly more analytical. It also describes the sequence of sounds (temporal order) and potential scene differences across audios. We refer to this as the tier-2 explanation.
**Detail**: This is the most detailed and descriptive answer. It delves into the sonic qualities of each audio, including potential sources and the listener's experience. It compares the two audios in terms of complexity and engagement, providing a richer and more immersive description. It analyzes the audio events, acoustic scenes, sound sources, signal characteristics, tonal differences, and overall feel of each audio. We refer to this as the tier-3 explanation.

The first-tier explanation is concise and to the point, the second-tier explanation provides more context, and the third explanation offers a thorough, immersive analysis. A secondary benefit of dividing the explanation into tiers allows us to do fine-grained analysis of model performance.

## 2.1 AUDIO DIFFERENCE DATASET

In this section, we outline the development steps for the AudioCaps Difference (ACD) and Clotho Difference (CLD) datasets.

**Audio recordings.** We source the audio recordings from the AudioCaps and ClothoV21 datasets. The AudioCaps dataset comprises 46,000 audio samples, each lasting 10 seconds, sourced from AudioSet, and includes human-annotated captions that describe the audio content. Annotators had access to both audio and visual cues during the annotation process. On the other hand, the Clotho dataset, though smaller, features audio samples ranging from 15 to 30 seconds, sourced from the Freesound platform, and includes five human-annotated captions per sample. These annotators only had access to audio information and followed a detailed protocol to minimize errors and ensure diversity in the captions. By utilizing both the AudioCaps and Clotho, we ensure variability in audio content, duration, and annotation style.

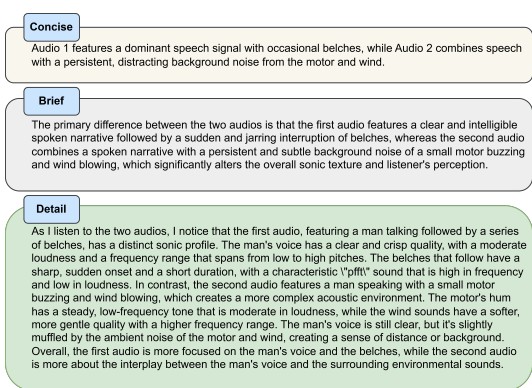

Figure 2: A random sample from the ACD dataset is displayed across three levels of explanation. The top pane provides a concise explanation, the middle pane offers a brief explanation, and the bottom pane presents a detailed explanation.

**Difference explanations.** Large Language Models (LLMs) have been effectively utilized to generate text descriptions for audio across various tasks, such as question-answering (Gong et al. (2024)), compositional reasoning (Ghosh et al. (2024b;a)), and deductive reasoning (Deshmukh et al. (2024c)). We adopt a similar approach for generating explanations for the audio difference task. The process involves three main steps: data sources, explanation generation, and explanation verification.

For data sources, we limit the audio recordings and human-annotated descriptions to those from AudioCaps and ClothoV21. To generate explanations, we prompt an LLM to describe the differences between two audio recordings using the provided human-annotated descriptions. Our prompting setup is similar to (Deshmukh et al. (2024c)), with two key modifications. First, we instruct the LLM to incorporate knowledge about sound sources, materials, acoustics, and emotions when generating explanations. Second, we define the tier of explanation by restricting the sources the LLM can use and the length of the explanation. Therefore, we sample two annotated human descriptions and prompt LLM to generate different tiers of explanations. Finally, human annotators verify the

difference explanations. If an explanation is found to be inaccurate, the human annotators manually remove the hallucinated audio event and add necessary details. Due to the cost of this process, we restrict verification to the test set of three tiers. Details on prompting setup and data creation are available in the Appendix E. The resulting dataset comprises of audio recordings paired with the three tiers of generated explanations.

| Data | Split | Examples per tier | Tier 1 | | | Tier 2 | | | Tier 3 | | |
|------|-------|-------------------|--------|------|--------|--------|------|--------|--------|------|--------|
| | | | Med. | Max | Vocab. | Med. | Max | Vocab. | Med. | Max | Vocab. |
| CLD | Train | 19195 | 27 | 49 | 6528 | 51 | 92 | 6462 | 155 | 221 | 10818 |
| CLD | Val | 5225 | 27 | 46 | 3743 | 51 | 89 | 4024 | 154 | 223 | 7026 |
| CLD | Test | 5225 | 27 | 28 | 5225 | 52 | 86 | 4059 | 156 | 219 | 7152 |
| ACD | Train | 48660 | 29 | 47 | 3287 | 52 | 104 | 8483 | 155 | 235 | 12891 |
| ACD | Val | 2456 | 28 | 46 | 2350 | 52 | 87 | 2563 | 154 | 227 | 4566 |
| ACD | Test | 4680 | 29 | 47 | 3287 | 53 | 95 | 3329 | 154 | 220 | 5489 |

Table 1: Dataset statistics of AudioCaps Difference (ACD) and Clotho Difference (CLD) dataset

**Dataset statistics.** The Audio difference dataset consists of $\{a_i, a_j, e_{ij}\}$ where $a_i$ is first audio, $a_j$ is the second audio, and $e_{ij}$ is explanation belonging to one of the three tiers. The statistics of AudioCaps Difference (ACD) and Clotho Difference (CLD) datasets across three tiers: Train, Validation, and Test splits are presented in Table 1. For example, in Tier 1, the ACD Train split has 48k examples with explanations having a median of 27, a maximum of 49, and a vocabulary size of 6528, while the CLD Train split has 19k examples and the explanations have a median of 51, a maximum of 92, and a vocabulary size of 6462. A randomly sampled example is shown in Figure 2.

## 2.2 EVALUATION

For objective evaluation, we use audio captioning metrics- BLEU (Papineni et al. (2002)), METEOR (Banerjee & Lavie (2005)), SPICE (Anderson et al. (2016)), CIDEr (Vedantam et al. (2015)), and SPIDEr (Liu et al. (2017)). BLEU measures n-gram precision, while METEOR combines precision and recall with features like stemming and synonymy matching. ROUGE focuses on recall of n-grams and sequences, CIDEr measures consensus in audio descriptions, and SPICE evaluates semantic content. SPIDEr combines SPICE and CIDEr for a balanced assessment, making it our primary metric for model comparison. More details on metric choice are available in Appendix I.1.

## 3 MODEL

In this section, we describe our proposed model ADIFF, which employs prefix tuning to prompt a frozen Language Model. The model accepts three inputs: audio 1, audio 2, and a user prompt, and generates free-form text as output. The model architecture is covered in Section 3.1, the training process in Section 3.2, the three stages of training in Section 3.3, and lastly inference in Section 3.4.

## 3.1 ARCHITECTURE

ADIFF consists of four main components: an audio encoder, an audio projection layer, a cross-projection layer, and a decoder-only language model.

**Audio Encoder.** The audio encoder is used for extracting general-purpose audio representations for both the audios. We use HTSAT (Chen et al. (2022)) which is pretrained on AudioSet (Gemmeke et al. (2017)) and achieves SoTA on identifying various sound events and acoustic scenes.

**Projection.** The audio projection is used to project the audio embeddings from audio encoder to latent space of Language model. The audio projection converts a single embedding into a sequence of latent tokens $[s, d]$. The projection first expands the hidden dimension to a larger hidden dimension $k$, which is then split to form $[s, d]$ where $k = s * d$. This is followed by concatenating with a learnable constant, resulting in $[s + c, d]$. This output is passed to the transformer, followed by clipping of the learnable constant output $c$. The resulting output of audio projection is of shape $[s, d]$. This projection architecture is shown to perform well for prefix-tuning architectures (Deshmukh et al. (2023a; 2024b); Mokady et al. (2021)). The text projection translates text embeddings with learnable transformer layers. The architecture is similar to audio projection (Fig 3) without the first linear layer.

**Cross Projection.** Once the audio embeddings are in the same latent space of Language model, the cross-projection layer is used to improve the model's ability to highlight differences. The cross-

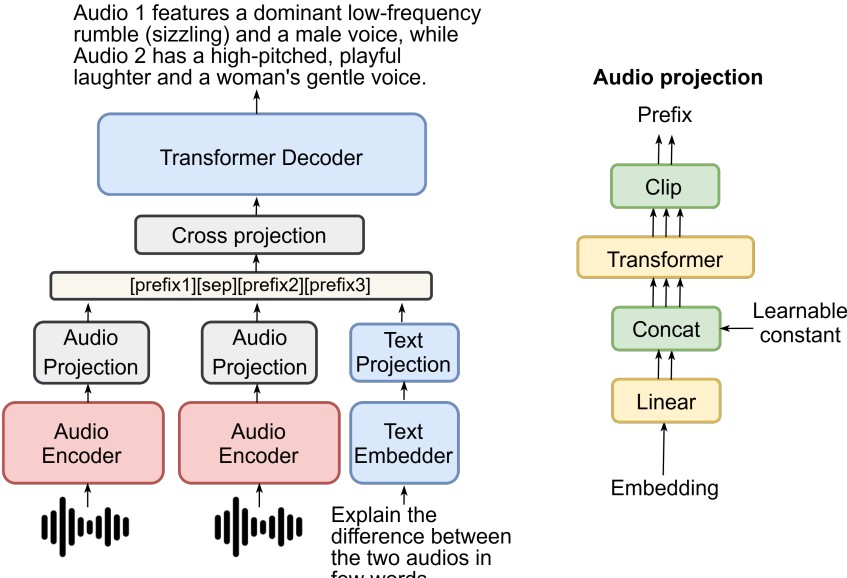

Figure 3: ADIFF takes two audio recordings and text prompt as input and generates free-form text as output. The two audios and prompt are independently encoded by the audio encoder and text embedder respectively, followed by projection layers to project the embeddings to the latent space of the transformer decoder. The two audio latent are separated by a separator token in latent space. The prefix formed by audio latent 1, SEP, audio latent 2 and text prompt prefix is fed to the cross-projection layer. The output of the cross-projection layer is used to prompt the transformer decoder to generate natural language explanations.

projection layer adds a separator token in latent space between the two audio embeddings followed by transformer layer to learn differences.

**Language Model.** The language model is a decoder-only language model which is used to auto-regressively generate text conditioned on the output of cross-projection and user prompt. We use GPT2 as the language model following prefix tuning literature (Deshmukh et al. (2023a; 2024b); Mokady et al. (2021)) and due to compute limitations.

### 3.2 TRAINING

The training process uses the next-token prediction objective to learn unfrozen (learnable) parameters. A sample input to the model is $\{x_1^i, x_2^i, t^i, c^i\}$ where $x_1^i$ is the first audio, $x_2^i$ is the second audio, $t^i$ is the text prompt, and $c^i$ is the difference explanation respectively. The user provides Audio 1 ($x_1^i$) and Audio 2 ($x_2^i$) along with a prompt ($t^i$) specifying the desired level of explanation. Each audio file is independently encoded by the audio encoder ($a_\phi$), creating separate audio embeddings. Simultaneously, the user's text prompt and separator token ($s$) is tokenized using a BPE tokenizer and embedded with the language model's vocabulary embeddings ($g_\psi$). These embeddings including the audios, separator embedding, and text prompt embeddings are then projected into the language model's latent space using a cross-projection layer ($m_\zeta$) layer to form a prefix.

$$c^i = c_1^i, ..., c_k^i = \text{concat}\{m_\zeta(a_\phi(x_1^i)), g_\psi(s), m_\zeta(a_\phi(x_2^i)), g_\psi(t^i)\} \tag{1}$$

$$p_1^i, ..., p_k^i = h_\beta(c^i) \tag{2}$$

The combined prefix $\{p_j^i\}_{j=1}^k$ is of length (tokens) $k$ in the language models space. This prefix is used to prompt the language model ($f_\theta$) to generate a text explanation highlighting the differences between the two audios. The model is trained as a typical captioning system, learning to predict a caption (text tokens) $o^i$ based on the prefix $p^i$ in an autoregressive manner. The loss function is Cross-Entropy per token:

$$\mathcal{L} = -\sum_{i=1}^{N}\sum_{j=1}^{l} \log p_\gamma(o_j^i | p_1^i, ..., p_k^i, o_1^i, ..., o_{j-1}^i) \tag{3}$$

where $\gamma$ denotes model's trainable parameters and consists of $\phi, \zeta, \psi, \beta$. Out of all the parameters- $\zeta, \beta$ are always trained while the rest are determined per stage (Section 3.3).

## 3.3 TRAINING STAGES

The model's training process involves several stages to preserve audio information and prevent the destructive gradient updates during unaligned full finetuning. These stages are: unimodal pretraining, multimodal alignment training, and finetuning.

**Unimodal Pretraining.** Unimodal pretraining entails training the audio encoder and language model on tasks specific to their respective modalities. For the audio encoder, this means training on AudioSet to predict audio events. For the language model, it involves pretraining on a large corpus of text. For the audio difference task, we skip independent modality pretraining and instead use pretrained unimodal models like HTSAT and GPT-2 which have performed the unimodal training.

**Multimodal grouding.** After unimodal pretraining, the language model must be grounded in the audio to generate audio-conditioned text. During this stage, both the audio encoder and language model are frozen, and the model learns the audio projection and cross projection using cross-entropy loss for each predicted next text token. Freezing the audio encoder and language model ensures that the initial gradient updates do not cause the loss of modality-specific information. This approach is similar to the prefix-tuning literature, where the language model remains frozen (Deshmukh et al. (2023a); Mokady et al. (2021)).

**Finetuning.** In this final stage, we finetune the audio-grounded language model by unfreezing all the modules. This allows us to retain audio-specific information and better steer the language model to produce the necessary descriptive answers using acoustic terminology. This stage uses a low learning rate with warmup and minimum training steps to minimize catastrophic forgetting.

## 3.4 INFERENCE

During inference, the prefix is created from the two test audios and the user-provided text prompt. The decoder (language model), generates the next token in sequence based on this prefix. At each step, the model assigns probabilities to all vocabulary tokens, which are then used to select the next token according to the chosen decoding method. In our experiments, we employed top-k and top-p decoding for better performance across all experiments. The details on experimental setup, training, inference, and implementation can be found in Appendix G.

| Task | Models | Tier 1 | | | Tier 2 | | | Tier 3 | | |
|------|--------|--------|--------|--------|--------|--------|--------|--------|--------|--------|
| | | BLEU$_4$ | METEOR | SPIDEr | BLEU$_4$ | METEOR | SPIDEr | BLEU$_4$ | METEOR | SPIDEr |
| ACD | Baseline | 0.118 | 0.210 | 0.220 | 0.163 | 0.193 | 0.225 | 0.153 | 0.188 | 0.123 |
| | QwenAC (L) | 0.132 | 0.214 | 0.235 | 0.166 | **0.204** | 0.212 | 0.165 | 0.202 | 0.173 |
| | QwenAC (F) | 0.110 | 0.183 | 0.258 | 0.163 | 0.199 | 0.241 | 0.151 | 0.194 | 0.082 |
| | ADIFF | **0.135** | **0.221** | **0.303** | **0.180** | 0.197 | **0.345** | **0.171** | **0.208** | **0.183** |
| CLD | Baseline | 0.128 | 0.237 | 0.212 | 0.233 | 0.234 | 0.641 | 0.157 | 0.199 | 0.166 |
| | QwenAC (L) | 0.140 | 0.285 | 0.230 | 0.232 | 0.236 | 0.756 | 0.155 | 0.200 | 0.182 |
| | QwenAC (F) | 0.126 | 0.232 | 0.204 | **0.273** | **0.254** | **0.958** | 0.130 | 0.207 | 0.172 |
| | ADIFF | **0.203** | **0.302** | **0.652** | 0.213 | 0.235 | 0.692 | **0.191** | **0.220** | **0.417** |

Table 2: Benchmarking different models on Audio Difference task. The top half of the table shows the performance of different models on the task of ACD, while The bottom half of the table shows the performance of different models on the task of CLD. All the models are trained on ACD and CLD train split. The Tier 1, 2, and 3 classifications correspond to the explanation tiers detailed in Section 2. The results with all metrics and the average score are available in Appendix Table 15.

## 4 RESULTS

### 4.1 EXPERIMENTAL SETUP

The base model includes an audio encoder, audio projection, and a language model (GPT2-base). It is trained end-to-end with the language model frozen, using captioning loss. This is referred to as the baseline or naive model. For ADIFF, we add a separator and a cross-projection layer to the architecture. ADIFF's training process has three stages to preserve audio information and prevent destructive gradient updates during unaligned full finetuning: unimodal pretraining, multimodal alignment training, and finetuning, as detailed in Section 3.3. For SoTA ALMs, we use Qwen-AC (Chu et al. (2023)) as it is the only ALM in literature that supports two audio inputs. We consider three versions of Qwen-AC: Zero-Shot (Z), LoRA (L), and full fine-tuning (F). All models except for the zero-shot version use the train set of ACD and CLD for training and are evaluated on the test set of ACD and CLD across three tiers.

| Model | LLM | Studio | | | FSD50K | | | GTZAN | | | Average | | |
|---|---|---|---|---|---|---|---|---|---|---|---|---|---|
| | | COR | GRA | RDB | COR | GRA | RDB | COR | GRA | RDB | COR | GRA | RDB |
| QwenAC (Z) | 7B | 2.73 | 2.64 | 3.09 | 2.76 | 2.20 | 3.28 | 2.95 | 2.83 | 3.33 | 2.81 | 2.56 | 3.23 |
| Baseline | 128M | 2.99 | 3.26 | 3.21 | 3.43 | 3.36 | 3.47 | 3.37 | 3.28 | 3.49 | 3.26 | 3.35 | 3.39 |
| QwenAC (L) | 7B | 3.05 | 3.41 | 3.25 | 3.35 | **3.73** | 3.45 | 3.31 | 3.29 | 3.63 | 3.24 | 3.48 | 3.44 |
| QwenAC (F) | 7B | 3.09 | 3.50 | **3.37** | 3.41 | 3.54 | 3.49 | 3.35 | 3.32 | 3.61 | 3.28 | 3.45 | 3.49 |
| ADIFF | 128M | **3.12** | **3.73** | 3.34 | **3.82** | 3.51 | **3.61** | **3.46** | **3.34** | **3.75** | **3.47** | **3.53** | **3.57** |

Table 3: Human evaluation for generated explanations based on Correctness (COR), Granularity (GRN), and Readability (RDB). For QwenAC, Z indicates zero-shot, L indicates LoRA finetune, and F indicates full-finetuning.

## 4.2 RESULTS

**Objective evaluation.** The performance of the naive model, QwenAC versions and ADIFF model is compared in Table 2, using objective metrics described in Section 2.2. The ADIFF model outperforms the naive baseline across all three tiers. Moreover, it also outperforms QwenAC (L) and QwenAC (F) across all Tiers of ACD and CLD except on Tier 2 CLD. For Tier 2, we observe similar trends as in ablation conducted for baseline architecture and language-only performance 5.1. For Tier 2, we observe similar trends to those seen in the ablation studies conducted for the baseline architecture and language-only performance. Linguistically, Tier 2 is the easiest to learn from, followed by Tier 3, and then Tier 1. This is because Tier 1 has the fewest words, with most of them containing audio-related information. In contrast, about 15% of the words in Tier 2 pertain to the linguistic structure of contrasting audios. This indicates that Tier 2's higher scores, even in subsequent experiments, are due to its linguistic simplicity for the model to match. Consequently, QwenAC, which uses a much stronger 7B parameter LLM, performs better than ADIFF, which has a 128M parameter LLM.

**Subjective evaluation.** The objective metrics are limited because they rely on linguistic biases, which can unfairly penalize diverse outputs (Morato & Mesaros (2021); Mei et al. (2024a)). Therefore, we also perform subjective evaluation, where human annotators rate the audio difference explanations across different dimensions of correctness (1-5), granularity (1-5), and readability (1-5). The dimension definitions and human evaluation setup and is explained in Appendix I.2. The subjective evaluation also ensures a fair comparison with QwenAC (Z), which is not trained on the ACD and CLD datasets and thus do not align with the data distribution, leading to poorer objective metric scores. We evaluate the models across three scenarios. The first scenario, Studio, involves recordings from the same sources, such as different breeds and numbers of dog barks, all recorded in a professional studio. The second scenario involves random samples from FSD50K (Fonseca et al. (2021)). The third scenario uses random samples from GTZAN Genres (Tzanetakis & Cook (2002)), featuring music from various genres and settings. The subjective evaluation results are shown in Table 3. Across all scenarios, ADIFF outperforms the naive baseline and Qwen-AC (Z) across all metrics and domains. Specifically, ADIFF sees the largest improvement in the granularity metric, highlighting the model's ability to produce in-depth descriptions. ADIFF also beats the fine-tuned variants of Qwen-AC on average across all metrics. However, in two cases—readability on Studio and granularity on FSD50K—a fine-tuned version of Qwen-AC outperforms ADIFF. This is likely because Qwen-AC's LLM, Qwen-7B, is significantly larger than the 128M LLM of ADIFF. Qualitative analysis shows it produces more coherent output, slightly improving readability. Additionally, we highlight the difficulty of the task, as even ADIFF achieves average scores of 3.5 across the dimensions out of 5.

## 5 ABLATIONS

In this section, we examine various components of model architecture and the impact of training methods on audio difference task performance. It covers- baseline architecture and the contribution of linguistics (Section 5.1), effect of cross-projection layer (Section 5.2), scaling language model (Section 5.3), position captioning (Section 5.4), and impact of stage three fine-tuning (Section 5.5).

## 5.1 BASELINE ARCHITECTURE

To build the baseline architecture for the audio difference task, we draw on previous works on prefix tuning (Deshmukh et al. (2023a)). This architecture includes an audio encoder to extract features, an audio mapper to translate these features into the latent space of a language model, and a frozen language model (GPT2-base) to generate text. For the audio difference task, we use the audio

encoder to extract features from both audio inputs, followed by mappers to convert these features into the language model's latent space. The frozen language model prompting and the cross-entropy loss setup remain unchanged. We refer to this as the baseline architecture for this task.

In question-answering and captioning tasks, models have been observed to learn linguistic information and answer questions without relying on perception modalities like vision or audio. This phenomenon is noted in both vision and audio literature (Lipping et al. (2022)). Similarly, for the audio difference task, models can game metrics by learning the language patterns instead of analyzing the audio, thereby inflating the metrics. To address this, we establish a baseline where the audio encoder is randomly initialized and kept frozen. This approach allows us to determine the maximum performance achievable without analyzing audio, focusing solely on learning the language patterns for each tier of explanation. We refer to this as the language-only performance for the audio difference task.

We summarize the language-only and baseline performance in Table 4. The full table is available in Appendix Table 16. Experiment A presents language-only performance, where the audio encoder is randomly initialized and frozen. In contrast, Experiment B shows baseline performance with the audio encoder initialized from HTSAT pretrained weights and frozen. In Experiment A, we observe that Tier 2 is the easiest to learn from a linguistic standpoint, followed by Tier 3, and then Tier 1. This is because Tier 1 has the fewest words, with most containing audio-related information. Conversely, about 15% of the words in Tier 2 pertain to the linguistic structure of contrasting audios. This suggests that Tier 2's higher scores, even in subsequent experiments, are due to its linguistic simplicity for the model to match. In Experiment B, where the audio encoder is pretrained, the model performs better across all tiers, as expected. This indicates that the model architecture effectively leverages audio information to improve explanations.

| Task | Exp. | Tier 1 | | | Tier 2 | | | Tier 3 | | |
|------|------|--------|--------|--------|--------|--------|--------|--------|--------|--------|
| | | $BLEU_4$ | METEOR | SPIDEr | $BLEU_4$ | METEOR | SPIDEr | $BLEU_4$ | METEOR | SPIDEr |
| ACD | A | 0.082 | 0.184 | 0.128 | 0.135 | 0.176 | 0.154 | 0.149 | 0.175 | 0.107 |
| | B | 0.118 | 0.210 | 0.220 | **0.163** | 0.193 | 0.225 | **0.199** | 0.188 | 0.123 |
| | C | **0.131** | **0.214** | **0.287** | 0.155 | **0.197** | **0.243** | 0.149 | **0.203** | **0.154** |
| CLD | A | 0.138 | 0.223 | 0.127 | 0.199 | 0.236 | 0.589 | 0.123 | 0.166 | 0.127 |
| | B | 0.128 | 0.237 | **0.212** | 0.233 | 0.234 | 0.641 | **0.157** | **0.199** | 0.166 |
| | C | **0.156** | **0.257** | 0.195 | **0.280** | **0.267** | **0.904** | 0.127 | 0.187 | **0.196** |

Table 4: Architecture results. Experiment A is baseline architecture with random audio encoder weights. Experiment B is pretrained audio encoder weights. Experiment C is ADIFF which modifies baseline architecture with separator token and cross-projection. The results with all metrics and the average score are available in Appendix Table 16

## 5.2 EFFECT OF CROSS-PROJECTION

We observe two limitations with the baseline model performance by qualitative analysis. First, when the audio input includes perceptually similar sounds and the same audio events, the model tends to confuse the distinct information between audios. Second, the model struggles with Tier 2 and Tier 3 explanations, where it needs to discern subtle audio differences such as scenes, acoustics, sources, and their composition rather than just audio events. To address the second limitation and partially the first, we introduce a cross-projection layer with a latent separator token between the two audio inputs. We concatenate the first audio latent embedding, the separator embedding, and the second audio latent embedding to form a prefix. The separator embedding is derived from the $< |endoftext| >$ token embedding of GPT-2 base. This concatenated prefix is then passed through the cross-projection layer, which consists of transformer layers with a learnable constant vector.

The results with the cross-projection layer are summarized in Table 4. The full table is available in Appendix Table 16. Experiment B shows the performance of baseline architecture. Experiment C shows performance of baseline architecture with separator token and cross-projection layer. On average, we see improvements for ACD and CLD across the three tiers of explanations. Specifically, in Appendix Table 16, the average metric score of Experiment C is consistently higher than Experiment B across all tiers of ACD and CLD.

## 5.3 SCALING LANGUAGE MODEL PARAMETERS

The performance of vision-language models (Alabdulmohsin et al. (2024)) tends to improve as their scale increases. Empirical evidence suggests that this improvement often follows a predictable

| Task | Exp. | Tier 1 | | | Tier 2 | | | Tier 3 | | |
|------|------|--------|--------|--------|--------|--------|--------|--------|--------|--------|
| | | BLEU$_4$ | METEOR | SPIDEr | BLEU$_4$ | METEOR | SPIDEr | BLEU$_4$ | METEOR | SPIDEr |
| ACD | Base | **0.131** | 0.214 | 0.287 | 0.155 | 0.197 | 0.243 | 0.149 | 0.203 | 0.154 |
| | Med | 0.119 | 0.211 | **0.325** | 0.154 | 0.198 | **0.248** | 0.160 | 0.201 | **0.232** |
| | Large | 0.124 | **0.221** | 0.240 | 0.156 | 0.197 | 0.147 | **0.162** | 0.207 | 0.165 |
| | XL | 0.129 | 0.216 | 0.272 | **0.164** | **0.202** | 0.215 | 0.154 | **0.208** | 0.107 |
| CLD | Base | **0.156** | 0.257 | 0.195 | **0.280** | 0.267 | **0.904** | 0.127 | 0.187 | 0.196 |
| | Med. | 0.149 | **0.270** | 0.317 | 0.264 | **0.287** | 0.641 | 0.143 | 0.203 | 0.204 |
| | Large | 0.133 | 0.238 | 0.263 | 0.180 | 0.223 | 0.574 | 0.177 | 0.191 | **0.227** |
| | XL | 0.112 | 0.256 | 0.275 | 0.234 | 0.255 | 0.527 | **0.146** | **0.208** | 0.212 |

Table 5: Scaling language model results. The language model in ADIFF architecture (transformer decoder) ranges in size from 128 million (Base) to 1.5 billion parameters (XL). The results with all metrics and the average score are available in Appendix Table 17

power law (Kaplan et al. (2020)). Similarly, recent studies have found a correlation between the scale of language models and their performance on audio reasoning tasks (Deshmukh et al. (2024c)). However, these observations are typically made at the compute-optimal frontier (Hoffmann et al. (2022)) and are not often examined under limited compute conditions. In practical scenarios, where compute resources and time are constrained, larger models may not be the best option. This has been illustrated in the case of language models, where the newer Chinchilla model (Hoffmann et al. (2022)) outperformed its predecessor Gopher (Rae et al. (2021)), despite being four times smaller. Therefore, we study scaling language for audio-language models for prefix tuning.

To investigate the impact of scale, we modify the language model (transformer decoder) in the architecture depicted in Figure 3. We experiment with GPT2-base (128M), GPT2-medium (256M), GPT2-large (774M), and GPT2-XL (1.5B). Each model is trained with the same compute budget, approximately equivalent to 30 epochs. The results, presented in Table 5, indicate that with the same data (audio and text tokens) and limited compute, the base and medium models perform similarly on average (Table 17), while the large and XL models perform worse. Moreover, the higher per-epoch computational cost of larger models widens the gap between small and large LM in ADIFF. It is important to note that we do not scale the number of training tokens as we scale the language models, which affects performance.

To determine if increased computational power helps larger LMs, we train the models longer for an additional 20 epochs. The outcomes, shown in Figure 4, indicate that for the 128M and 256M models, performance peaks around 30 epochs. In contrast, for the 774M and 1.5B models, performance continues to improve with more epochs. This suggests that aligning and guiding larger models with prefix-tuning requires a greater number of epochs. Given the computation budget and performance, we choose to use GPT2-base for ADIFF and subsequent experiments.

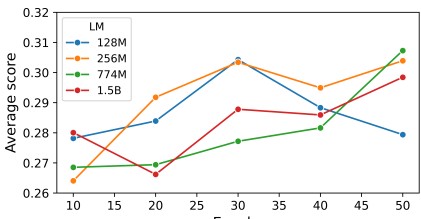

Figure 4: Change in average score across tiers with increase in LM parameters.

### 5.4 AUDIO GROUNDING WITH POSITION CAPTIONING

To improve audio grounding, we instruct the model to caption either audio 1 or audio 2. This is accomplished by training the model on both audio captioning data and audio difference datasets. The audio captioning data includes {audio1, audio2, prompt}, where the prompt is "caption the first audio" or "caption the second audio". By incorporating single or position-specific captioning data in the training process, we ensure the model does not get confused between audio 1 and audio 2, and accurately distinguishes between similar or acoustically alike audio events.

Table 6 presents the results after incorporating audio captioning data. Experiment D uses baseline architecture with cross-projection, while the bottom section shows the same architecture enhanced with captioning data. For reference, Experiment C in Table 4 is the performance of baseline architecture with cross-projection. On average, the ACD dataset shows improvements across all Tiers, whereas the CLD dataset yields mixed results. This discrepancy can be attributed to AudioCaps being sourced from AudioSet, which mainly contains speech recordings. As a result, the ACD dataset includes more instances of human speech being compared under various acoustic conditions. In contrast, the CLD dataset features a wider variety of audio comparisons. Despite the mixed per-

formance, we continue to use the model trained with audio captioning, as it demonstrates greater improvements during stage-3 finetuning.

| Task | Exp. | Tier 1 | | | Tier 2 | | | Tier 3 | | |
|------|------|--------|--------|--------|--------|--------|--------|--------|--------|--------|
| | | BLEU$_4$ | METEOR | SPIDEr | BLEU$_4$ | METEOR | SPIDEr | BLEU$_4$ | METEOR | SPIDEr |
| ACD | D | 0.129 | 0.208 | 0.300 | 0.162 | 0.195 | 0.296 | 0.164 | 0.199 | 0.150 |
| | E | **0.135** | **0.221** | **0.303** | **0.180** | **0.197** | **0.345** | **0.171** | **0.208** | **0.183** |
| CLD | D | 0.191 | **0.304** | 0.511 | **0.236** | **0.242** | 0.689 | 0.136 | 0.188 | 0.124 |
| | E | **0.203** | 0.302 | **0.652** | 0.213 | 0.235 | **0.692** | **0.191** | **0.220** | **0.417** |

Table 6: Audio grounding and finetuning results. Experiment D is the ADIFF model with position captioning. Experiment E finetunes the language model of ADIFF along with position captioning. The results with all metrics and the average score are available in Appendix Table 18

## 5.5 STAGE-3 FINETUNING

In the final stage, following multimodal grounding, we fine-tune the language model. This training is conducted over a few epochs with a small learning rate of approximately 1e-6, using a cosine learning rate scheduler. The results of this fine-tuning are presented in Table 6. Experiment D is position captioning training, while Experiment E shows the performance post-fine-tuning. The final fine-tuning consistently improves performance across various tiers and datasets. We also find that the finetuned model outperforms the base model across three tiers, especially Tier 3. More details on qualitative analysis are available in Appendix K.

## 6 HALLUCINATION

Language models often generate misleading or incorrect outputs, known as hallucinations. For Audio-Language Models (ALMs), this means producing responses not based on actual audio input, such as inventing audio events or misinterpreting common sounds (Kuan et al. (2024); Nishimura et al. (2024)). This lack of grounding can impair deductive reasoning and comparative skills. To address this, we use audio grounding with position captioning to differentiate similar-sounding events from different sources. However, some hallucination issues persist. We aim to provide tools to detect these hallucinations by keeping the HTSAT audio encoder frozen. This encoder, trained on AudioSet, predicts 527 audio events and their presence probabilities over time, allowing users to verify the accuracy of generated descriptions.

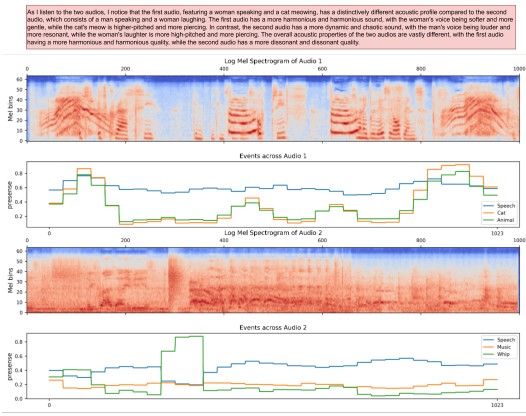

Figure 5: Audio event presence probabilities from AD-IFF to detect hallucinations.

An example of this is shown in Figure 5. The top pane displays the audio difference explanation generated by the model. For each audio, the log mel spectrogram and the top three audio event presence probabilities over time are plotted. From the figure, we can see that the model missed the whip sound in audio 2 in the difference explanation and provides a way of analyzing explanations.

## 7 CONCLUSION

In this paper, we introduce the task of audio difference explanation, create relevant datasets, and propose an strong baseline. We present two new datasets from AudioCaps and Clotho. Using Large Language Models (LLMs), we generated three levels of explanations for audio differences: concise descriptions, brief sentences about events and scenes, and comprehensive explanations including semantics and emotions. We propose ADIFF, an audio prefix tuning-based language model with a cross-projection module and a three-step training process to address the limitation of the naive baseline. The objective and subjective evaluations show significant performance improvements over the naive baseline and SoTA ALM. We also conduct ablation studies to understand the effects of various components and find various insights relevant to training audio-language models. Our benchmarks and ablation studies show the effectiveness of our method, utilizing comparative reasoning to generate more human-like explanations for audio differences.

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

## A    ETHICS STATEMENT

This paper adheres to the ethical guidelines set forth by the ICLR conference and aims to contribute to the research community in a responsible and transparent manner. The study utilizes publicly available datasets, namely AudioCaps (Kim et al. (2019)) and Clotho (Drossos et al. (2020)), which have been used in accordance with their respective licenses and ethical use guidelines. No personally identifiable information (PII) or sensitive data was used in this research, ensuring the privacy and confidentiality of all subjects involved.

The research focuses on developing methods for explaining audio differences using natural language, and all experiments were conducted with the intent to advance scientific knowledge in a positive and constructive way. We acknowledge the potential implications of our work in fields such as audio forensics and surveillance, and we emphasize that our models are intended solely for academic and benign applications. We do not condone or support the misuse of our work in ways that could infringe upon individual privacy or be used for unethical purposes. All human evaluators and annotators participated voluntarily with their informed consent. There were no conflicts of interest or biases influencing the results of this study. Finally, we are committed to open science and will release our datasets and model checkpoints, to the research community to promote transparency, reproducibility, and collaboration.

## B    REPRODUCIBILITY STATEMENT

To ensure the reproducibility of our work, we have taken several measures, which are thoroughly documented throughout the main paper, appendix, and supplemental materials. For our proposed model, ADIFF, and the associated benchmarks, we have provided a downloadable source code anonymously in the supplementary materials to facilitate replication of our experiments. Detailed explanations of our experimental setup, hyperparameters, and model architecture can be found in Section 3 and Appendix G. We have outlined the data processing steps, including the creation of the AudioCaps Difference (ACD) and Clotho Difference (CLD) datasets, in Section 2.1 and the Appendix E, ensuring transparency in data preparation and usage. All datasets used in this work are publicly available, and we have provided links to these resources along with data processing scripts in the supplementary materials. Our ablation studies in Section 5 further validate the robustness of our model design, and comprehensive results are presented to demonstrate consistency and reliability of the findings.

## C    RELATED WORK

**Audio-text learning.** Audio-text learning focuses on developing multi-modal representations to address specific tasks. These tasks include text-to-audio retrieval (Deshmukh et al. (2023b)), audio captioning (Drossos et al. (2020); Mei et al. (2022a)), and text-to-audio generation (Liu et al. (2023)). Among these, audio captioning is particularly relevant for the task of audio difference explanation, as it involves generating natural language descriptions from audio recordings. The architecture typically comprises an encoder to learn audio representations and a decoder to generate text tokens (Drossos et al. (2017)). Research in this area has explored enhancing diversity (Morato & Mesaros (2021); Mei et al. (2024a; 2022b)) in outputs, improving audio encoder and training methods (Wu et al.), and leveraging LLMs as decoder (Tang et al. (2024a)).

**Audio-Language Models.** Instead of building task-specific models, recent literature has focused on building general-purpose audio models by using language. These models are referred to as Audio-Language Models. For example, CLAP (Elizalde et al. (2023; 2024); Wu et al. (2023)) and AudioFlamingo (Kong et al. (2024)) is trained on millions of audio-text pairs to learn multi-modal audio-text representations, and are used for tasks like zero-shot and few-shot classification and retrieval. To address open-ended tasks such as audio question answering and audio captioning, a decoder or LLM is added to the architecture. For example, Pengi (Deshmukh et al. (2023a)) employed prefix-tuning to learn a mapper and prompt a frozen language model. The later Audio-Language Models like LTU (Gong et al. (2024; 2023)), SALMONN (Tang et al. (2024b)), GAMA (Ghosh et al. (2024a)) utilized LoRA (Hu et al.) to develop lightweight adapters on language models. While most Audio Language Models (ALMs) concentrate on non-speech audio, Qwen-audio and LTU-AS integrate both speech and audio understanding by utilizing the Whisper encoder (Radford et al. (2023)).

**Difference Captioning.** In the vision domain, the task of explaining the difference between two images is studied under Image Difference Captioning (IDC) (Jhamtani & Berg-Kirkpatrick (2018); Park et al. (2019)). The architectures explored include encoder-decoder (Jhamtani & Berg-Kirkpatrick (2018)), attention-based (Park et al. (2019)), clip-based decoder (Guo et al. (2022)), and cross-modal transformer (Yao et al. (2022)). In the audio literature, the first relevant work Komatsu et al. (2024) explored using difference learning to improve audio captioning. The data consists of a mixture with audio 1 and audio 2, and then the model predicts the caption for audio 1, indirectly highlighting the difference. This approach differs from directly comparing two audios and highlighting differences; instead, it focuses on captioning one of the audios. The second work (Takeuchi et al. (2023)) focuses on audio difference captioning, aiming to find the semantic difference between similar but slightly different audio clips. The difference captioning takes the form of "change," "remove," or "add", for example, "change sound of footsteps." The task is concerned with adding or removing something from audio 1, instead of explaining how the audios differ concerning audio events, sound sources, acoustics, and emotional response. Therefore, in the audio literature, the task of audio difference explanation with different tiers of explanation is yet to be explored.

## D    COMPARATIVE REASONING FOR AUDIO

Comparative reasoning (Yu et al. (2023)) for audio involves analyzing and contrasting different audio sources, events, and signals to draw meaningful conclusions. This process typically starts with identifying the audio sources, which could range from live recordings, synthesized sounds, or pre-recorded tracks. Each source has unique characteristics such as frequency range, amplitude, and timbre. Audio events, which are specific occurrences within the audio signal like a note being played or a word being spoken, are then identified and analyzed. Temporal characteristics, including the timing, duration, and sequence of these events, play a crucial role in comparative reasoning. By examining these elements, one can discern patterns, anomalies, and relationships within the audio data. The signal processing aspect of comparative reasoning involves breaking down the audio signal into its fundamental components. For example, analysis of the spectral content, Signal-to-noise ratio (SNR), harmonic distortion, and other metrics can be used to evaluate the quality and integrity of the audio signal. By comparing these metrics across different audio sources or events, one can make informed judgments about their similarities and differences. Moreover, Comparative reasoning in audio is closely related to deductive reasoning (Johnson-Laird (1999)), where conclusions are drawn based on the analysis of specific data points. For instance, if two audio signals share similar temporal and spectral characteristics, one might deduce that they originate from the same source or have been processed similarly. Conversely, significant differences in these characteristics could indicate different sources or processing methods. By systematically comparing and contrasting audio data, one can make logical inferences and draw conclusions that are supported by empirical evidence. Therefore, enabling ALMs to comparative reasoning improves their overall logical reasoning ability.

## E    USING LLMS TO GENERATE AUDIO DIFFERENCE EXPLANATION

Large Language Models (LLMs) have been effectively utilized to generate text descriptions for audio across various tasks, such as question-answering, compositional reasoning, and deductive reasoning. These models leverage vast amounts of data and sophisticated algorithms to understand and generate human-like text. We adopt a similar approach for generating explanations for audio difference tasks, which involve identifying and describing differences between two audio recordings. The process involves three main steps: data sources, explanation generation, and explanation verification.

**Data Sources:** For our study, we limit the audio recordings and human-annotated descriptions to those from two audio captioning datasets: AudioCaps and ClothoV21. AudioCaps is a large-scale dataset that provides audio recordings along with detailed human-annotated descriptions. ClothoV21 is another comprehensive dataset that includes a variety of audio recordings with corresponding textual descriptions. By using these datasets, we ensure that our audio samples and descriptions are diverse and representative of real-world scenarios.

**Explanation Generation:** To generate explanations, we prompt an LLM to describe the differences between two audio recordings using the provided human-annotated descriptions. To select captions for prompting LLM, we perform random sampling per audio to eliminate the edge case of multiple captions. This process involves three steps. First, we flatten the .csv files to include one audio file and a single caption. For Clotho, which has 5 captions, this results in 5 entries per audio. Second,

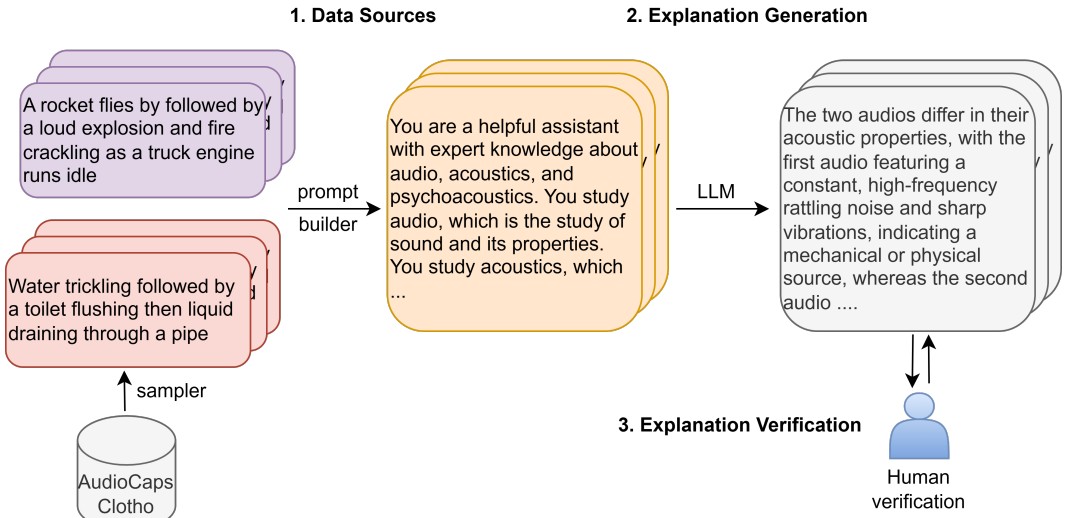

Figure 6: Using LLM to generate audio difference explanation. The process involves three key steps: sampling from human-annotated data sources, generating explanations, and verifying the generated explanations. The verification step is conducted only for the test set.

---

**Prompt for LLM**

You are a helpful assistant with expert knowledge about audio, acoustics, and psychoacoustics. You study audio, which is the study of sound and its properties. You study acoustics, which revolve around the generation, propagation, and reception of sound waves. You study Psychology which posits that a sound is a complex stimulus that encompasses a vast range of acoustic properties involving aspects of cognition, psychoacoustics, and psychomechanics. Your task is to perform audio captioning which consists of describing audio content using natural language. To describe the acoustic content, you utilize words related to their acoustic properties, such as their semantic relations, their spectro-temporal characteristics, frequency, loudness, duration, materials, interactions, and sound sources.

---

Table 7: Base prompt for LLM before generating responses

we conduct sampling. For instance, if the dataset contains N audios, we loop through the N audios and, for any $i^{\text{th}}$ audio, randomly pick a second audio ($j^{\text{th}}$) not including the i to i+4th audio. This is achieved by constructing an index list that excludes the i, i+1, ..., i+4 indices and sampling from this list. Third, the selected $i^{\text{th}}$ and $j^{\text{th}}$ audios are then used to look up the captions for the $i^{\text{th}}$ and randomly picked $j^{\text{th}}$ audio. These captions then become the seed for the LLM to generate the difference explanation. Our LLM prompting setup is similar to the one used in previous studies Deshmukh et al. (2024c), with two key modifications. First, we instruct the LLM to incorporate knowledge about sound sources, materials, acoustics, and emotions when generating explanations. This ensures that the explanations are not only accurate but also rich in context and detail. Second, we define the tier of explanation by restricting the sources the LLM can use and the length of the explanation. This tiered approach allows us to generate explanations of varying complexity and detail. The base prompt is detailed in Table 7.

**Explanation Verification**: Finally, human annotators verify the difference explanations generated by the LLM. If an explanation is found to be inaccurate, the human annotators manually remove the hallucinated audio event and add necessary details to correct it. Due to the cost and effort involved in this process, we restrict verification to the test set of three tiers. This ensures that our testing dataset is both accurate and reliable. The resulting dataset comprises audio recordings paired with three tiers of generated explanations.

## F ANALYSIS OF GENERATED DESCRIPTION

The three tiers of audio difference explanation are explained in Section 2, while the dataset creation process is explained in Section 2.1 and Section E. In creating the dataset, we rely on LLM to generate

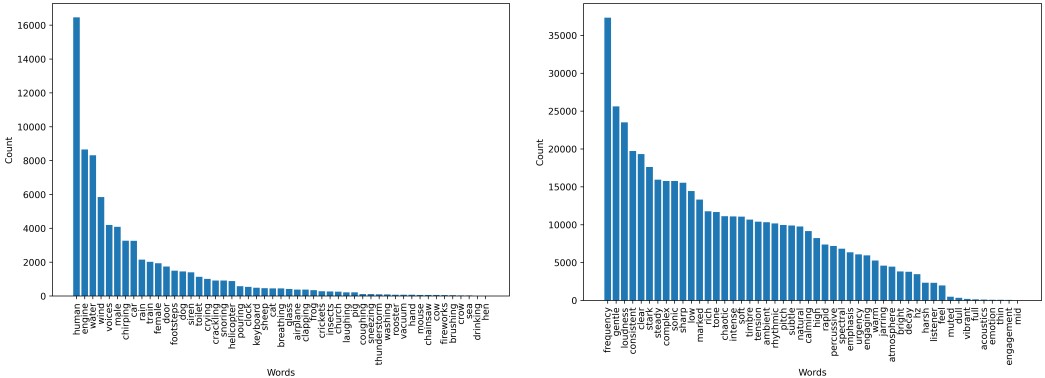

Figure 7: Comparison of vocabulary differences between explanatory tiers: The left figure illustrates words that are present in the Tier 2 explanation but not in the Tier 1 explanation, while the right figure highlights words unique to the Tier 3 explanation that are absent in the Tier 2 explanation.

the base difference explanation before it is verified for human annotators for the test-set. Therefore, it's important to explore the types of audio differences described in different tiers to uncover insights and variations in the three tiers of difference explanations. For instance, by splitting text into words, converting them to lowercase, and focusing on words related to audio, we can identify words that appear in Tier 2 but not in Tier 1, showing unique audio characteristics used for the difference explanation in Tier 2. Similarly, we can use the same method to examine the distinctions between Tier 2 and Tier 3, where the differences often lie in more detailed and immersive sound characteristic descriptions. The results, presented in Figure 7, highlight two main points. First, the unique words between Tier 3 and Tier 2 are significantly more than those between Tier 2 and Tier 1. This is expected, as Tier 3 explanations add considerably more detail than Tier 2. Second, the left plot of Figure 7 shows that the additional characteristics mentioned in Tier 2 over Tier 1 focus on audio events, sound sources, and the nature of the sounds, making it more analytical. In contrast, the right plot of Figure 7 illustrates that the additional characteristics in Tier 3 over Tier 2 focus on the sonic qualities of each audio, including potential sources, listener's experience, comparing the two audios in terms of the complexity and engagement, and therefore providing a more comprehensive difference description. Overall, we see the audio characteristics used in Tier 3 over Tier 2 (Figure 7, right subplot) and Tier 2 over Tier 1 (Figure 7, left subplot) are different, and focus on the different aspect and details between two audios. We further analyze the density of information across three Tiers in Appendix Section Q.

## G  EXPERIMENTAL SETUP

**Encoder and Decoder.**  We utilized the HTSAT (Chen et al. (2022)) audio transformer, trained on AudioSet (Gemmeke et al. (2017)), as our audio encoder. The audio was sampled at 32 kHz and converted into log Mel spectrograms with 64 Mel bins, a hop size of 320 ms, and a window size of 1024 ms. For HTSAT, all audio files were randomly truncated to 10 seconds in length. To maintain computational efficiency, the maximum length of user text input was set to 40. The audio encoder remained frozen throughout the experiments, enabling hallucination output using audio event detection probabilities. For the decoder or Language model, we use the GPT2 series of models, specifically the GPT2-base (124M) variant. The model remained unchanged throughout all the experiments except for Section (5.3).

**Projection.**  The audio projection uses an 8-layer transformer with a prefix length of 40. When combining two audio inputs, the total prefix length becomes 80. Adding a separator token (1 token) and a text projection (40 tokens) results in a total prefix length of 121. This combined input is then processed by the cross-projection layer, a 4-layer transformer, which maintains the prefix length of 121 in its output.

**Training.**  We use Adam Optimiser (Kingma & Ba (2015)) with warmup and step decay of the learning rate. For stage 2, multimodal grounding, all models are trained for 30 epochs on 8 A100 GPUs. Only for Section (5.3), we retrain the models for 50 epochs. In the final stage of fine-

tuning, we limit the training to 10 epochs to prevent the training loss from plateauing and to avoid catastrophic forgetting.

**Inference.** We use top-k and top-p decoding because of its better performance in producing diverse explanations that accurately capture details. We explore different decoding strategies and compare the performance of top-k and top-p decoding against the greedy decoding in Section N.

## H    USER TEXT PROMPT

Audio-based prefix-tuning models have been utilized for various tasks (Deshmukh et al. (2024b); Bukhari et al. (2024)), including as supervision to learn general-purpose audio representations (Deshmukh et al. (2023a)). The dataset for training these models consists of (audio, user input, output) triplets. During training, the user input remains fixed for a specific task and only varies across tasks. Consequently, the model performs the necessary task only when the exact text prompt or user input is provided. This limitation restricts the model's applicability. To address this, we explore supporting multiple input text prompts by randomly sampling prompts during training.

For the audio difference task, the ADIFF model must produce three tiers of explanation based on the user prompt. Additionally, it needs to generate audio captions for the first or second audio to enhance grounding (5.4). We aim for the user to be able to request different levels of explanation using flexible prompting, and the model should still provide accurate outputs. To achieve this, we create a database of user prompts for each tier. During training, depending on the required output tier, we sample a specific prompt from the database. In our case, the database is a simple dictionary containing about 100 sample user inputs per output tier. Randomly sampled examples of the user inputs are shown in Table 8.

| Tier | Randomly sampled user prompts |
|------|-------------------------------|
| 1 | "Summarize the differences between the two audios briefly.", "Highlight the main differences between the two audio tracks.", "Provide a concise comparison of the two audios." |
| 2 | "Explain the contrast between the two audio pieces in one long sentence.", "In one elaborate sentence, summarize the differences between both audio files.", "Describe the variation between the two audio tracks in one extended sentence." |
| 3 | "Explain the difference between both audios in detail.", "Could you elaborate on the distinctions between the two audio tracks?", "Please provide a detailed comparison of both audio files." |

Table 8: Randomly sampled three user prompts for each tier of audio difference explanation.

## I    EVALUATION

### I.1    OBJECTIVE EVALUATION

For objective evaluation, we leverage audio captioning metrics: BLEU, METEOR, SPICE, CIDEr, SPIDEr. The BLEU (Bilingual Evaluation Understudy) metric measures the precision of n-grams between the generated text and reference text, making it simple and fast, though it doesn't account for recall and can be insensitive to word order and meaning. METEOR (Metric for Evaluation of Translation with Explicit ORdering) combines precision and recall, incorporating features like stemming and synonymy matching, which results in better correlation with human judgment at the sentence level, albeit with increased complexity and computational demands. ROUGE (Recall-Oriented Understudy for Gisting Evaluation) focuses on the recall of n-grams and longest common subsequences, making it popular for text summarization, though its recall-centric nature might not always reflect overall quality. CIDEr (Consensus-based Image Description Evaluation) measures consensus between generated image descriptions and multiple reference descriptions, excelling in image captioning but requiring multiple reference captions for accuracy. SPICE (Semantic Propositional Image Caption Evaluation) evaluates the semantic content of image captions by comparing scene graphs, capturing semantic meaning more effectively than surface-level metrics, though it is computationally intensive. Lastly, SPIDEr (SPICE and CIDEr) combines SPICE and CIDEr to leverage both semantic and consensus-based evaluation, providing a balanced assessment but inheriting the computational complexity of both metrics. Therefore, we primarily use SPIDEr to compare models or calculate the average score across all the above metrics for model comparison.

## I.2 SUBJECTIVE EVALUATION

While objective metrics are correlated with human evaluation, they have their limitations. Therefore, we also set up a human evaluation task to assess the difference explanations generated by the model. In this task, evaluators are asked to evaluate the differences between two audio samples. These differences could pertain to various aspects such as audio events (e.g., specific sounds or occurrences within the audio), acoustic scenes (e.g., the overall environment or context of the audio), signal characteristics (e.g., frequency, amplitude, timbre), and listener emotions (e.g., the emotional impact or response elicited by the audio). Evaluators then provide scores based on three dimensions: Correctness, Granularity, and Readability. These dimensions ensure that the evaluations are comprehensive, focusing on the accuracy, detail, clarity, and acoustic quality of the explanations provided. The evaluation dimensions and the corresponding score descriptions are:

**Correctness (1-5)**

- 1: The explanation is mostly incorrect, with significant inaccuracies in identifying audio events, acoustic scenes, or signal characteristics.

- 2: The explanation has several inaccuracies but captures some correct audio events and acoustic scenes.

- 3: The explanation is partially correct, with a mix of accurate and inaccurate details.

- 4: The explanation is mostly correct, with minor inaccuracies.

- 5: The explanation is entirely correct, accurately identifying all relevant audio events, acoustic scenes, and signal characteristics.

**Granularity (1-5)**

- 1: The explanation is very vague, lacking detail and specificity.

- 2: The explanation provides some detail but is generally too broad.

- 3: The explanation is moderately detailed, covering key aspects but missing finer points.

- 4: The explanation is detailed, covering most aspects with good specificity.

- 5: The explanation is highly detailed, thoroughly covering all relevant aspects with precise specificity.

**Readability (1-5)**

- 1: The explanation is very difficult to read, with poor structure and clarity.

- 2: The explanation is somewhat difficult to read, with several issues in structure and clarity.

- 3: The explanation is moderately readable, with some issues in structure and clarity.

- 4: The explanation is mostly readable, with minor issues in structure and clarity.

- 5: The explanation is very easy to read, with excellent structure and clarity.

We recruited five professionals to evaluate the generated audio difference explanations. Each annotator was asked to use headphones to listen to both audio samples. After listening and analyzing the two audios, annotators provided scores based on three criteria: correctness, granularity, and readability. Each criterion had a maximum score of 5. The exact task guidelines are shown in Table 9, an individual instance of the task is shown in Table 10, and the results of the human evaluation are presented in Table 3.

## J  CROSS-PROJECTION

We introduce a cross-projection layer with a latent separator token between two audio inputs to address two key challenges. First, it enhances the model's ability to differentiate between perceptually similar sounds and identical audio events. Second, it improves Tier 2 and Tier 3 explanations, which involve discerning subtle audio differences such as scenes, acoustic sources, and their composition, rather than merely identifying audio events. From an architectural standpoint, we concatenate the first audio latent embedding, the separator embedding, and the second audio latent embedding to form a prefix. The separator embedding is derived from the $<$ —endoftext— $>$ token embedding of GPT-2. This concatenated prefix is then processed through the cross-projection layer, which consists of transformer layers with a learnable constant vector. The output of the cross-projection layer is in the same latent space as GPT-2, allowing us to analyze the cross-projection outputs.

**Task Instructions**

Your task is to rate the difference explanation between the two audios. First, listen to each audio clip and analyze the difference between the two audios. Pay attention to differences in terms of audio events (dog barking vs cat meowing), sound sources (musical instrument, human voice), acoustic scene (park vs living room), signal characteristics (frequency, pitch, loudness, etc) emotion response (listening to the audio invokes a sense of happiness, fear, in you) and any other difference using world-knowledge or using deductive reasoning. Once you have analyzed the clip, asses the below audio difference explanations provided.

**Important Notice:**

Please note that certain sections of this audio test may include unexpectedly loud sounds. To ensure a comfortable experience, we suggest setting your volume to a suitable level before starting and adjusting it as necessary throughout the test.

Table 9: Task guidelines provided to annotators.

**Please read the task guidelines before completing the task. First, listen to the two audio recordings carefully and analyze the difference between them. Second, read the descriptions which explains the difference between the two audios. Finally, provide your rating of the difference explanation across three dimensions of correctness, granularity and readability**

**How accurate is the audio difference explanation?**

**1 (Poor)** The explanation is mostly incorrect, with significant inaccuracies in identifying audio events, acoustic scenes, or signal characteristics.
**2 (Fair)** The explanation has several inaccuracies but captures some correct audio events and acoustic scenes.
**3 (Good)** The explanation is partially correct, with a mix of accurate and inaccurate details.
**4 (Very Good)** The explanation is mostly correct, with minor inaccuracies.
**5 (Excellent)** The explanation is entirely correct, accurately identifying all relevant audio events, acoustic scenes, and signal characteristics.

**How detailed and granular is the audio difference explanation?**

**1 (Poor)** The explanation is very vague, lacking detail and specificity.
**2 (Fair)** The explanation provides some detail but is generally too broad.
**3 (Good)** The explanation is moderately detailed, covering key aspects but missing finer points.
**4 (Very Good)** The explanation is detailed, covering most aspects with good specificity.
**5 (Excellent)** The explanation is highly detailed, thoroughly covering all relevant aspects with precise specificity.

**How easy to read and understand is the audio difference explanation?**

**1 (Poor)** The explanation is very difficult to read, with poor structure and clarity.
**2 (Fair)** The explanation is somewhat difficult to read, with several issues in structure and clarity.
**3 (Good)** The explanation is moderately readable, with some issues in structure and clarity.
**4 (Very Good)** The explanation is mostly readable, with minor issues in structure and clarity.
**5 (Excellent)** The explanation is very easy to read, with excellent structure and clarity.

Table 10: Individual labeling instance for audio difference explanation task.

Let the output of the cross-projection layer be [n, d], where (n) represents the number of tokens, equal to the prefix length (121), and (d) is the hidden dimension, matching GPT-2's intermediate layer size of 768. The GPT-2 vocabulary embedding has dimensions [v, d], where (v) is the vocabulary size and (d) is the hidden dimension. To find the closest match between the cross-projection tokens (n) and the vocabulary embeddings (v), we normalize the vectors and compute the dot product between them. The highest dot product indicates the most likely vocabulary token. These vocabulary tokens are numerical and tokenized using the BPE tokenizer. We then detokenize them to obtain English subwords. As expected, the interpreted prefix initially appears as gibberish, similar to findings in prefix tuning for other tasks. However, these gibberish English words can be parsed to extract meaningful audio words or subwords. Below, we present these subwords for readability.

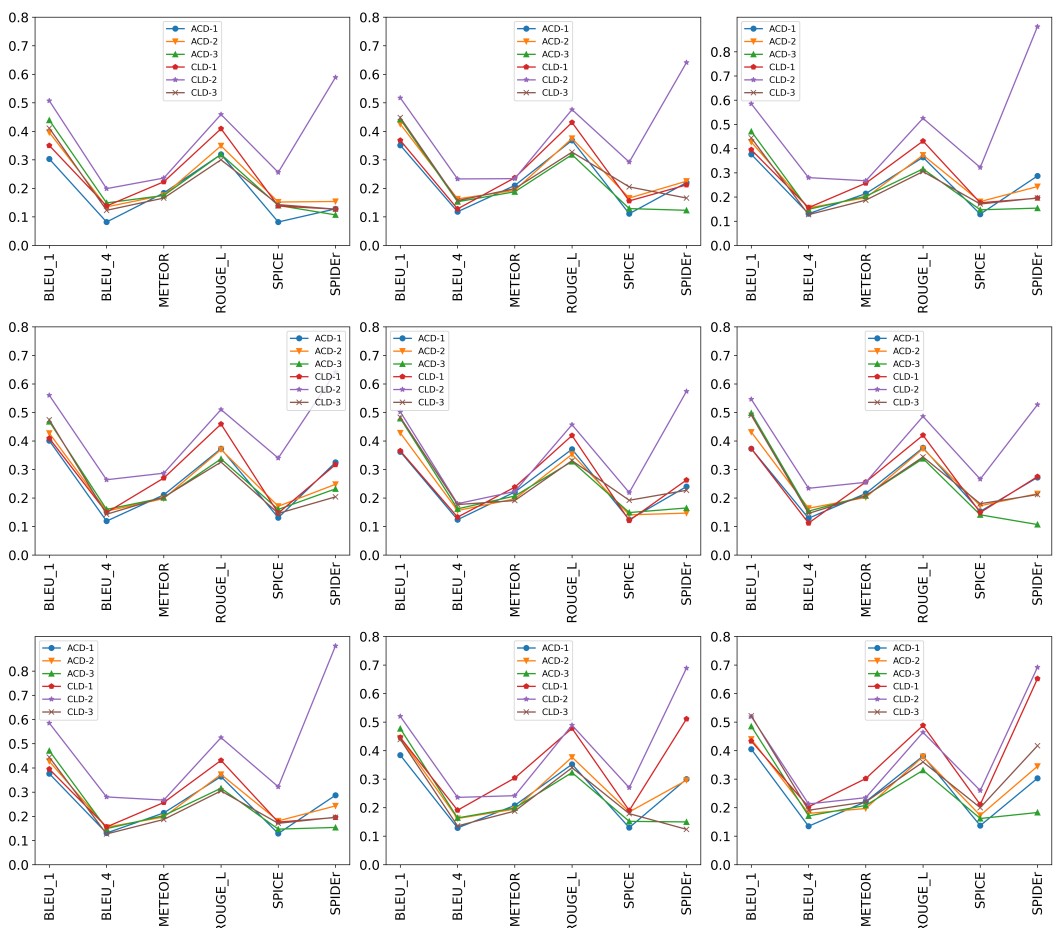

Figure 8: The plots for different ablation studies are organized as follows: The (1,1) subplot represents the naive baseline with random audio encoder weights, while the (1,2) subplot shows the naive baseline with HTSAT pretrained audio encoder weights. The (1,3) subplot features ADIFF with cross-projection and GPT2-base. Moving to the second row, the (2,1) subplot illustrates ADIFF with the GPT2-med language model, the (2,2) subplot displays ADIFF with the GPT2-large language model, and the (2,3) subplot presents ADIFF with GPT2-XL. In the third row, the (3,1) subplot replicates the ADIFF result from subplot (1,3), the (3,2) subplot shows ADIFF with audio position training, and the (3,3) subplot combines ADIFF with audio position training and a finetuning stage.

| Audio 1 prefix | Audio 2 prefix | Text prefix |
|---|---|---|
| explosion, crack, hit | helicop, loud | mid-high, frequency, pitch, dynamic, range |
| bee, robots voices, repetition, downstairs, speakers | motor, rustic, radio pursuit, vintage throttle | development, plus, nature, catch, dynamic range, loud, single |
| baby, cry, speaker, female | speaker, cheering, male | clean, predict, loud, greater |
| clap, man, speaker, after | drum, bang, tik | dur, loud, intelli, soft, surprise |

Table 11: Parsed words and subwords from the interpreted prefix. The prefix here is the output of the cross-projection layer and the corresponding model outputs are shown in Table 12

We observe that even after the cross-projection layer is supposed to mix information between audios, the prefix for audio 1 retains information about audio 1, while the prefix for audio 2 retains information about audio 2. Instead, by using the cross-projection layer, it reuses the text prefix to store attributes and difference information, which is later used to form the explanation. Therefore, the text prefix gets used for not only steering the language model but also for storing information about attributes to be used for comparison. For example, in the first row of Table 11, the text prefix contains mid-range, frequency, and pitch attributes, which differ between the two audios and appear in the ADIFF output (Table 12, row 1). Similarly, in the second example, we see loudness and dynamic range attributes in the prefix, with corresponding difference explanations using those

for comparison (Table 12, row 2). The main caveat of these results is that we are approximately estimating the English token, assuming the output of the cross-projection is in the same space as GPT-2 and using a dot product. However, this result still provides insight into the model's workings and how the cross-projection layer benefits and improves model performance for Tier 2 and Tier 3 explanations.

# K  EFFECT OF STAGE-3 FINETUNING

We also perform qualitative analysis and find that Model 2 consistently outperformed Model 1 across three tiers. For example, one randomly picked example of which is shown in Figure 11. For Tier 1, Model 2 accurately described the ground truth for both audio samples, identifying the mix of human voices, animal sounds, and environmental noise in Audio 1, and the presence of machine sounds and animal noises without human speech in Audio 2. In Tier 2, Model 2 captured the specific details of the dominant sound sources, correctly identifying the complexity and variety of sounds in Audio 1 and contrasting them with the mechanical sounds and horse's hooves in Audio 2. For Tier 3, Model 2 provided a more accurate and detailed comparison, describing the distinct sounds and their acoustic properties, as well as the overall ambiance and spatial characteristics of each audio, making it a more comprehensive and precise analysis compared to Model 1.

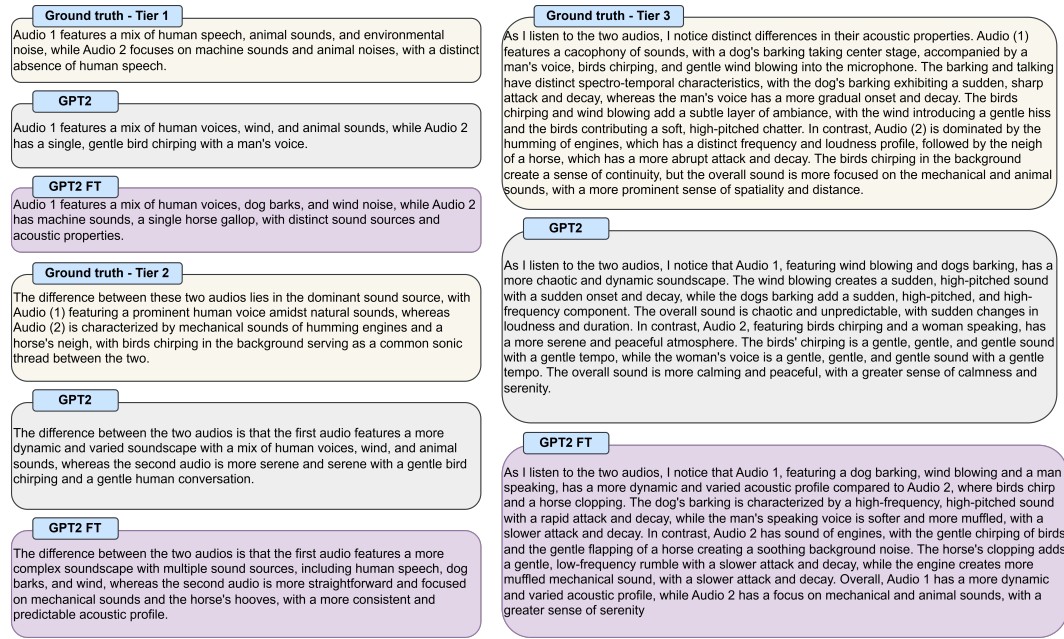

Figure 9: We compare two models: first, GPT2, the model after stage-2 multimodal grounding, Second, GPT2 FT after stage-3 finetuning. The architecture for both models is the same and consist of audio encoder, audio projection, cross-projection, and GPT2-base as the decoder. The top pane provides a concise explanation, the middle pane offers a brief explanation, and the bottom pane presents a detailed explanation.

# L  ABLATION EXTENSION

The ablations results are shown in Figure 8 for easy comparison. In the (1,1) subplot, the language-only model (with a randomly initialized and frozen audio encoder) demonstrates lower performance, particularly in Tier 1, which is the hardest due to its limited words and focus on audio-related content. In contrast, Tier 2 is the easiest to learn due to its simpler linguistic structure, and Tier 3 falls in between. The (2,1) subplot, which presents baseline model performance with a pretrained and frozen audio encoder, shows improved results across all tiers. However, qualitative analysis reveals two limitations with this baseline model: difficulty distinguishing between perceptually similar sounds and struggling to differentiate subtle audio differences in Tiers 2 and 3. To address these issues, a cross-projection layer with a latent separator token is introduced, as shown in the (3,1) subplot. This

adjustment improves performance, particularly in distinguishing audios in ACD and CLD datasets across all tiers.

Further experiments to explore the impact of model scale are depicted in subplots (2,1), (2,2), and (2,3). Using GPT2-base, GPT2-medium, and GPT2-large (256M, 774M, and 1.5B parameters, respectively), the models were trained with the same compute budget. Results show that the base and medium models perform similarly, while the larger models require more epochs to achieve better performance. The (3,2) subplot presents results after incorporating audio captioning data to further improve audio grounding, where the model distinguishes between audios more accurately, especially in the ACD dataset, which includes more human speech comparisons. However, the CLD dataset shows mixed results due to its wider variety of audio comparisons. Despite this, the audio captioning model continues to be used for stage-3 finetuning. Finally, the (3,3) subplot presents the performance after fine-tuning the language model over a few epochs with a small learning rate. This stage consistently improves performance across tiers and datasets, with the finetuned model outperforming the base model (subplot (3,1)) in all cases.

## M    QUALITATIVE ANALYSIS

We perform analysis of explanations generated by ADIFF. The different scenarios and explanation are shown in Table 12. In the first scenario, Audio (1) features a sudden, sharp explosion followed by crackling sounds, while Audio (2) presents a continuous low-frequency hum of a helicopter. This comparison highlights differences in frequency content and dynamic range, with Audio (1) having a more gradual onset and decay, and Audio (2) exhibiting sudden changes in loudness and frequency. The second scenario contrasts a baby crying and a woman speaking in Audio (1) with a man speaking and a crowd cheering in Audio (2). Here, the focus is on the distinct acoustic signatures and dynamic ranges, with Audio (1) being more consistent and predictable, while Audio (2) has greater variability in loudness and pitch. Here, we compare different voice characteristics which involves understanding pitch, frequency, and intensity. In the third scenario, the model compares a brief, loud applause followed by a male voice in Audio (1) with sharp, percussive sounds in Audio (2). This scenario emphasizes differences in acoustic properties such as frequency, loudness, and duration. The fourth scenario involves a dog barking loudly in Audio (1) and several dogs barking with a man speaking in Audio (2), highlighting the chaotic and dissonant quality of the latter. In this scenario, we compare model performance on similar audio events to spot differences. Finally, the fifth scenario compares a group playing a piano in Audio (1) with a piano followed by a drum beat in Audio (2), focusing on the timbre and frequency content. These scenarios were chosen to evaluate the model's ability to describe various audio events and scenes.

We also compare ADIFF generation against existing ALMs in the literature. As of yet, only Qwen-Audio Chat (Chu et al. (2023)) supports two audios as input. The qualitative comparison is shown in Figure 10. We consider three scenarios: Contrasting Acoustic Qualities, Overlapping Sound Sources, and Overlapping Sound Events. For Contrasting Acoustic Qualities, Qwen Audio identified the first clip as someone cutting wood with a chainsaw and the second as someone walking on snow, while ADIFF provided a detailed analysis, noting the harsh, high-frequency sound of the chainsaw and the mellow, low-frequency sound of walking on snow. In the Overlapping Sound Sources scenario, Qwen Audio described both clips as featuring dogs barking, with the first being playful and the second aggressive, whereas ADIFF detailed the first clip's consistent, sharp barks and the second clip's chaotic mix of dogs barking and a man speaking. Lastly, in the Overlapping Sound Events scenario, Qwen Audio simply stated that both clips were the same song performed by different orchestras, while ADIFF offered a detailed comparison, highlighting the rich, resonant timbre of the first clip's piano and the structured sound profile of the second clip's piano and drum beat, noting the first as harmonious and calming and the second as energetic and attention-grabbing. Overall, we find ADIFF explanations to be more accurate (audio event coverage) and with finer granularity compared to Qwen-Audio.

## N    GENERATION DIVERSITY

If we utilize greedy decoding for both inference and evaluation, it results in outputs with repetitive starting phrases like "As I listen to the two audios" and the same tokens being used to describe sound events and scenes, as shown in Table 12. This repetition is expected due to the nature of greedy decoding, which selects the most likely token at each step. To enhance the diversity of the generated

| Scenario | Difference explanation from ADIFF |
|---|---|
| 1 | As I listen to the two audios, I notice that Audio (1) features a sudden, sharp, and high-frequency explosion followed by a series of crackling sounds, likely generated by a small object being hit. The explosion's frequency content is likely dominated by mid-to-high frequencies, with a prominent presence of low-frequency energy. In contrast, Audio (2) presents a continuous, low-frequency hum of a helicopter flying overhead, with a consistent pitch and amplitude. The helicopter's sound is likely generated by a series of high-frequency transients, with a prominent presence of mid-to-high frequencies. The overall dynamic range of Audio (2) is much greater, with sudden changes in loudness and frequency content, whereas Audio (1) has a more gradual onset and decay. |
| 2 | As I listen to the two audios, I notice that the first audio, featuring a baby crying and a woman speaking, has a distinct acoustic signature. The baby's cries are characterized by high-pitched, irregular patterns of sounds, with varying frequencies and durations, creating a sense of urgency and distress. The woman's voice, on the other hand, is softer and more gentle, with a consistent pitch and volume. In contrast, the second audio, featuring a man speaking and a crowd cheering, has a more structured sound profile. The man's voice is clear and intelligible, with a consistent pitch and volume, while the crowd's cheering adds a layer of complexity with varying frequencies, loudness, and durations. The overall dynamic range of the second audio is much greater, with sudden changes in loudness and pitch, whereas the first audio has a more consistent and predictable loudness. |
| 3 | The two audios differ significantly in terms of their acoustic properties, with the first audio featuring a sudden, brief, and loud applause followed by a clear and intelligible male voice, whereas the second audio consists of a series of sharp, percussive, and percussive sounds from the tapping and banging, with distinct differences in frequency, loudness, and duration. |
| 4 | As I listen to the two audios, I notice that the first audio, featuring a dog barking loudly, has a distinct acoustic signature characterized by a consistent, repetitive pattern of barks with varying frequencies and durations. The sound is loud and has a sharp attack, with a relatively short duration. In contrast, the second audio, featuring several dogs barking and a man speaking, has a more chaotic and dissonant quality. The dog's barks are loud and have a wide range of frequencies, from low rumbles to high-pitched yips, with varying durations and intensities. The man's voice is softer and more muffled, with a more gradual attack and decay. The overall texture of the second audio is more chaotic and dissonant, with a greater emphasis on the dog's vocalizations and the man's speech. |
| 5 | I notice that the first audio, featuring a group of people playing a piano, has a distinct acoustic signature characterized by a rich, resonant timbre with a wide range of frequencies, from low rumbles to high-pitched notes. The sound is rich in low-frequency energy, with a prominent presence of mid-range frequencies, giving it a bright and piercing quality. In contrast, the second audio, featuring a piano playing followed by a drum beat, has a more structured and structured sound profile. The piano's timbre is characterized by a consistent pitch, with a prominent presence of high-frequency energy, creating a sense of rhythm and melody. The drum beat's frequency content is more balanced, with a prominent presence of low-frequency energy, creating a sense of tension and release. Overall, the first audio is more harmonious and calming, while the second audio is more energetic and attention-grabbing. |

Table 12: Example outputs generated from ADIFF for different scenarios

**Contrasting Acoustic Qualities**

**Qwen Audio**

Yes, there is a difference between the two audio clips. The first audio clip is of someone cutting wood with a chainsaw, while the second audio clip is of someone walking on snow.

**ADIFF**

As I listen to the two audios, I notice that the first audio, featuring a person cutting wood, has a distinctively harsh and abrasive quality to it. The sound is characterized by a high-frequency content, likely in the range of 2-5 kHz, which gives it a harsh and abrasive quality. The sound is also quite loud, with a duration that is quite long, suggesting a sustained and intense sound. In contrast, the second audio, featuring someone walking through snow, has a more mellow and gentle quality. The sound is marked by a lower frequency content, likely in the range of 100-200 Hz, which gives it a softer and more gentle quality. The sound is also quite loud, with a duration that is quite long, suggesting a sustained and intense sound. Overall, the two audios have distinct acoustic properties that reflect their respective sound sources and environments.

**Overlapping Sound Sources**

**Qwen Audio**

Both audio clips feature dogs barking. However, the first audio clip features dogs barking in a playful and excited manner, while the second audio clip features dogs barking in a more aggressive and anxious manner.

**ADIFF**

As I listen to the two audios, I notice that the first audio, featuring a dog barking loudly, has a distinct acoustic signature characterized by a consistent, repetitive pattern of barks with varying frequencies and durations. The sound is loud and has a sharp attack, with a relatively short duration. In contrast, the second audio, featuring several dogs barking and a man speaking, has a more chaotic and dissonant quality. The dogs' barks are loud and have a wide range of frequencies, from low rumbles to high-pitched yips, with varying durations and intensities. The man's voice is softer and more muffled, with a more gradual attack and decay. The overall texture of the second audio is more chaotic and dissonant, with a greater emphasis on the dog's vocalizations and the man's speech.

**Overlapping Sound Events**

**Qwen Audio**

The two audio clips are of the same song performed by different orchestras.

**ADIFF**

I notice that the first audio, featuring a group of people playing a piano, has a distinct acoustic signature characterized by a rich, resonant timbre with a wide range of frequencies, from low rumbles to high-pitched notes. The sound is rich in low-frequency energy, with a prominent presence of mid-range frequencies, giving it a bright and piercing quality. In contrast, the second audio, featuring a piano playing followed by a drum beat, has a more structured and structured sound profile. The piano's timbre is characterized by a consistent pitch, with a prominent presence of high-frequency energy, creating a sense of rhythm and melody. The drum beat's frequency content is more balanced, with a prominent presence of low-frequency energy, creating a sense of tension and release. Overall, the first audio is more harmonious and calming, while the second audio is more energetic and attention-grabbing.

Figure 10: Comparing ADIFF generation against current ALMs across three scenarios: Contrasting Acoustic Qualities, Overlapping Sound Sources, and Overlapping Sound Events. Notably, Qwen-Audio Chat is the only ALM in the literature that supports two audio inputs.

content, we explore top-k and top-p decoding. We set top-p to a high value (0.8) to limit the sampling of low-probability tokens. Additionally, we choose a top-k value of 3, allowing other high-scoring tokens a chance to be selected, thereby introducing randomness and improving the diversity of the output. The results are shown in Table 13. Using greedy decoding, the temporal order of events for audio 1 is incorrectly identified. However, with top-k and top-p decoding, the temporal order of events for audio 1 is correctly identified. Overall, top-k and top-p decoding increases the diversity

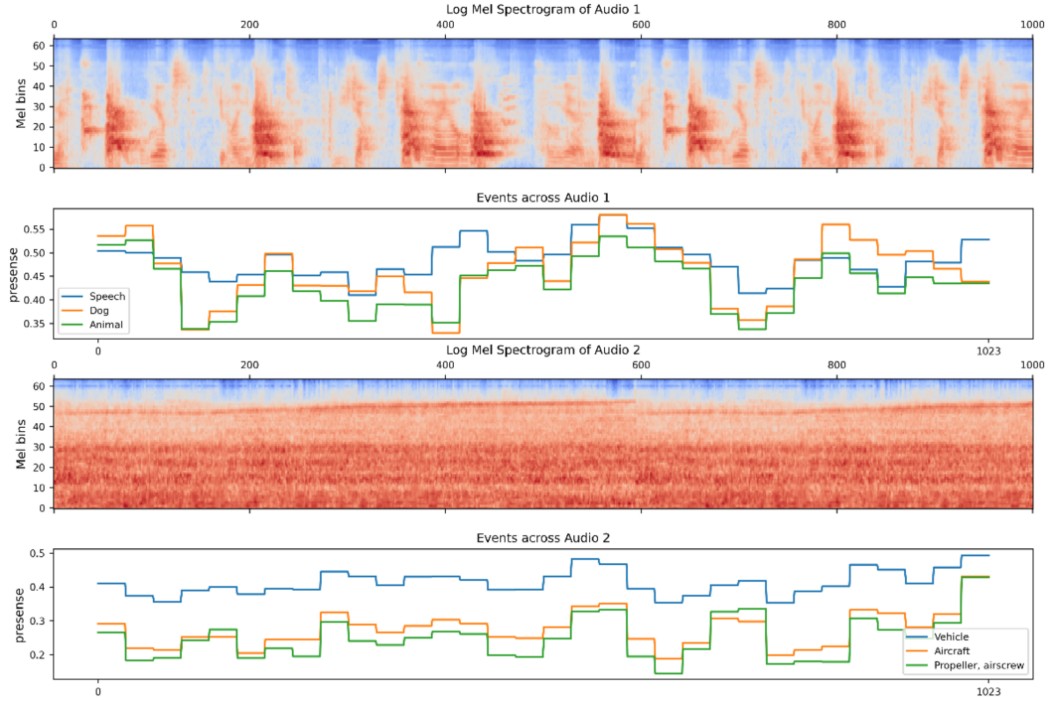

Figure 11: A randomly selected example showcases the model's detailed explanation of audio differences, accompanied by log mel spectrograms and audio presence plots. The top pane displays the model-generated explanation. For each audio clip, it includes the log mel spectrogram and the top three audio event presence probabilities over time.

of the output, reducing monotonicity and helps to improve performance. The results with top-k and top-p decoding are compared against greedy decoding in Table 25.

## O  ADIFF WITHOUT CROSS-PROJECTION LAYER

In Table 5.2, we examine the impact of the cross-projection layer. The middle section of the table presents the performance of the baseline architecture, while the bottom section illustrates the performance with the inclusion of the separator token and cross-projection layer. On average, we notice improvements for ACD and CLD across all three tiers of explanations. However, in the final ADIFF model, the LLM is fine-tuned, potentially learning what the cross-projection layer achieves, thereby rendering it redundant. To address this, we conducted another ablation study comparing the ADIFF model with three-step training without the cross-projection layer against the ADIFF model with the cross-projection layer. The results of this experiment are shown in Table 14. We found that the difference between ADIFF with and without the cross-projection layer is reduced. Nevertheless, ADIFF with the cross-projection layer still outperforms ADIFF without the cross-projection layer across all tiers for the ACD and CLD datasets. We hypothesize that when the LM is fine-tuned, the cross-projection layer acts as block attention over the audio and text prefixes, which has been shown to improve performance in vision LLMs Beyer et al. (2024). Thus, retaining the cross-projection layer still leads to better performance even when the LLM is fine-tuned.

## P  USING WAVCAPS FOR TRAINING

AudioCaps Kim et al. (2019) and Clotho Drossos et al. (2020) are used for audio captioning evaluation and used in every Audio-Language Model paper Deshmukh et al. (2023a); Ghosh et al. (2024a);

| Decoding | Generated difference explanation |
|---|---|
| Ground truth | As I listen to the two audios, I notice that the first audio is characterized by a sudden onset of sounds, starting with the clear and crisp voice of a young female, followed by a sharp, high-pitched sound of spraying, and then a piercing scream. In contrast, the second audio features a steady, gentle flow of water, accompanied by a soft and distant voice of a woman speaking. The most striking difference is the dynamic range and temporal structure of the two audios. The first audio has a sudden and dramatic change in sound levels, with a clear distinction between the soft voice and the loud spraying and screaming, whereas the second audio has a more consistent and gradual sound level, with the water flow and distant voice blending together seamlessly. |
| Greedy | As I listen to the two audios, I notice that Audio 1, featuring a girl speaking and spraying, has a distinctively different acoustic profile compared to Audio 2, which consists of water running and a distant woman speaking. The first audio is characterized by a sudden, sharp attack of a spray sound, followed by a woman's voice speaking in a calm tone, and then a sudden, sharp attack of a spray sound. The frequency content of the two audios is also distinct, with Audio 1 featuring a wider range of frequencies, from low rumbles to high-pitched squeals, whereas Audio 2 has a more even distribution of frequencies across the audio spectrum. Overall, the two audios have distinct acoustic properties, with Audio 1 featuring a sudden, attention-grabbing event and Audio 2 featuring a calming, serene atmosphere. |
| Top-k and Top-p (1) | As I listen to the two audios, I notice that Audio 1, featuring a woman talking followed by spraying and spraying noise, has a distinct sonic landscape. The woman's voice is characterized by a high-frequency range, likely in the 100-500 Hz range, with a rapid tempo and a steady loudness level. The spraying noise adds a sudden, high-energy event, followed by a sharp, transient event, and then a brief pause before a woman speaks again. In contrast, Audio 2, featuring water running, has a more gradual and continuous sound, with a gentle, mid-frequency range (500-2000 Hz) that provides a sense of texture and continuity. The sound of the water is characterized by a consistent, gentle flow, with a consistent loudness level and a steady tempo. The difference lies in the type of sound sources, materials, and interactions involved, with Audio 1 featuring sudden, attention-grabbing events and Audio 2 presenting a more continuous, soothing background. |
| Top-k and Top-p (2) | As I listen to the two audios, I notice a stark contrast between them. Audio (1) features a sudden, high-frequency burst of a female voice, followed by a sudden, sharp, and high-frequency spraying sound, and then another female voice. The audio is characterized by a sharp attack and decay, with a clear and intelligible voice that is likely in the mid-to-high frequency range. In contrast, Audio (2) features a gentle, low-frequency gurgling sound from a water source, accompanied by distant speech. The audio is dominated by mid-to-high frequency range, with a smooth and continuous quality that is characteristic of water sounds. The acoustic properties of these two audios are fundamentally different, with Audio (1) evoking a sense of excitement and excitement, while Audio (2) evokes a sense of serenity and relaxation. |

Table 13: Generated outputs from different decoding methods. (1) and (2) indicates run 1 and run 2 respectively.

| Task | Tier | BLEU$_1$ | BLEU$_2$ | BLEU$_3$ | BLEU$_4$ | METEOR | ROUGE$_L$ | CIDEr | SPICE | SPIDEr | AVG |
|------|------|-------|-------|-------|-------|--------|---------|-------|-------|--------|-----|
| ACD | 1 | 0.384 | 0.256 | 0.182 | 0.129 | 0.208 | 0.352 | 0.470 | 0.130 | 0.300 | 0.268 |
| ACD | 2 | 0.444 | 0.304 | 0.223 | 0.162 | 0.150 | 0.377 | 0.407 | 0.185 | 0.296 | 0.288 |
| ACD | 3 | 0.477 | 0.305 | 0.214 | 0.164 | 0.199 | 0.323 | 0.148 | 0.152 | 0.150 | 0.237 |
| CLD | 1 | 0.446 | 0.338 | 0.258 | 0.191 | 0.304 | 0.478 | 0.832 | 0.190 | 0.511 | 0.394 |
| CLD | 2 | 0.517 | 0.354 | 0.261 | 0.194 | 0.231 | 0.458 | 0.971 | 0.251 | 0.611 | 0.427 |
| CLD | 3 | 0.496 | 0.325 | 0.228 | 0.166 | 0.203 | 0.357 | 0.232 | 0.184 | 0.208 | 0.266 |
| ACD | 1 | 0.405 | 0.276 | 0.192 | 0.135 | 0.221 | 0.381 | 0.469 | 0.137 | 0.303 | 0.280 |
| ACD | 2 | 0.440 | 0.301 | 0.232 | 0.180 | 0.197 | 0.379 | 0.518 | 0.173 | 0.345 | 0.307 |
| ACD | 3 | 0.486 | 0.317 | 0.225 | 0.171 | 0.208 | 0.331 | 0.204 | 0.162 | 0.183 | 0.254 |
| CLD | 1 | 0.433 | 0.338 | 0.265 | 0.203 | 0.302 | 0.488 | 1.093 | 0.211 | 0.652 | 0.443 |
| CLD | 2 | 0.519 | 0.362 | 0.274 | 0.213 | 0.235 | 0.464 | 1.124 | 0.260 | 0.692 | 0.460 |
| CLD | 3 | 0.522 | 0.355 | 0.255 | 0.191 | 0.220 | 0.360 | 0.634 | 0.200 | 0.417 | 0.350 |

Table 14: ADIFF model performance without cross-projection layer. The top half of the table shows the performance of the ADIFF model without the cross-projection layer. The bottom half of the table shows the performance of the ADIFF model with the cross-projection layer. Both models are trained on ACD and CLD train split. The Tier 1, 2, and 3 classifications correspond to the explanation tiers detailed in Section 2.

| Task | Tier | BLEU$_1$ | BLEU$_2$ | BLEU$_3$ | BLEU$_4$ | METEOR | ROUGE$_L$ | CIDEr | SPICE | SPIDEr | AVG |
|------|------|-------|-------|-------|-------|--------|---------|-------|-------|--------|-----|
| ACD | 1 | 0.351 | 0.247 | 0.174 | 0.118 | 0.210 | 0.368 | 0.329 | 0.111 | 0.220 | 0.236 |
| ACD | 2 | 0.425 | 0.298 | 0.221 | 0.163 | 0.193 | 0.375 | 0.284 | 0.166 | 0.225 | 0.261 |
| ACD | 3 | 0.442 | 0.282 | 0.199 | 0.153 | 0.188 | 0.318 | 0.117 | 0.129 | 0.123 | 0.217 |
| CLD | 1 | 0.368 | 0.266 | 0.192 | 0.128 | 0.237 | 0.431 | 0.270 | 0.156 | 0.212 | 0.251 |
| CLD | 2 | 0.517 | 0.376 | 0.294 | 0.233 | 0.234 | 0.476 | 0.990 | 0.292 | 0.641 | 0.450 |
| CLD | 3 | 0.448 | 0.298 | 0.211 | 0.157 | 0.199 | 0.327 | 0.127 | 0.205 | 0.166 | 0.238 |
| ACD | 1 | 0.382 | 0.257 | 0.185 | 0.132 | 0.214 | 0.374 | 0.366 | 0.105 | 0.235 | 0.250 |
| ACD | 2 | 0.426 | 0.296 | 0.221 | 0.166 | 0.204 | 0.377 | 0.234 | 0.191 | 0.212 | 0.259 |
| ACD | 3 | 0.480 | 0.306 | 0.216 | 0.165 | 0.202 | 0.329 | 0.196 | 0.151 | 0.173 | 0.246 |
| CLD | 1 | 0.387 | 0.288 | 0.217 | 0.140 | 0.285 | 0.456 | 0.287 | 0.174 | 0.230 | 0.274 |
| CLD | 2 | 0.514 | 0.376 | 0.292 | 0.233 | 0.236 | 0.473 | 1.272 | 0.239 | 0.756 | 0.488 |
| CLD | 3 | 0.485 | 0.308 | 0.214 | 0.155 | 0.200 | 0.310 | 0.186 | 0.178 | 0.182 | 0.247 |
| ACD | 1 | 0.359 | 0.224 | 0.156 | 0.110 | 0.183 | 0.331 | 0.411 | 0.104 | 0.258 | 0.237 |
| ACD | 2 | 0.442 | 0.302 | 0.219 | 0.163 | 0.199 | 0.376 | 0.311 | 0.170 | 0.241 | 0.269 |
| ACD | 3 | 0.462 | 0.290 | 0.200 | 0.151 | 0.194 | 0.309 | 0.030 | 0.134 | 0.082 | 0.206 |
| CLD | 1 | 0.350 | 0.258 | 0.195 | 0.126 | 0.231 | 0.401 | 0.290 | 0.117 | 0.204 | 0.241 |
| CLD | 2 | 0.558 | 0.421 | 0.336 | 0.273 | 0.254 | 0.517 | 1.620 | 0.301 | 0.958 | 0.581 |
| CLD | 3 | 0.468 | 0.282 | 0.185 | 0.130 | 0.207 | 0.328 | 0.162 | 0.182 | 0.172 | 0.235 |
| ACD | 1 | 0.405 | 0.276 | 0.192 | 0.135 | 0.221 | 0.381 | 0.469 | 0.137 | 0.303 | 0.280 |
| ACD | 2 | 0.440 | 0.301 | 0.232 | 0.180 | 0.197 | 0.379 | 0.518 | 0.173 | 0.345 | 0.307 |
| ACD | 3 | 0.486 | 0.317 | 0.225 | 0.171 | 0.208 | 0.331 | 0.204 | 0.162 | 0.183 | 0.254 |
| CLD | 1 | 0.433 | 0.338 | 0.265 | 0.203 | 0.302 | 0.488 | 1.093 | 0.211 | 0.652 | 0.443 |
| CLD | 2 | 0.519 | 0.362 | 0.274 | 0.213 | 0.235 | 0.464 | 1.124 | 0.260 | 0.692 | 0.460 |
| CLD | 3 | 0.522 | 0.355 | 0.255 | 0.191 | 0.220 | 0.360 | 0.634 | 0.200 | 0.417 | 0.350 |

Table 15: Comparing baseline models against ADIFF model performance. The top half of the table shows the performance of the naive model. The second part shows the performance of QwenAC finetuned with LoRA. The third part shows the performance of QwenAC finetuned fully. While the bottom half of the table shows the performance of the ADIFF model. Both models are trained on ACD and CLD train split. The Tier 1, 2, and 3 classifications correspond to the explanation tiers detailed in Section 2.

Gong et al. (2024; 2023); Chu et al. (2023); Tang et al. (2024b) in literature. Therefore, we use AudioCaps Kim et al. (2019) and Clotho Drossos et al. (2020) for creating the ACD and CLD dataset. The human annotations in AudioCaps and Clotho may have problems, therefore we also perform human verification on the test difference explanations generated to remove any discrepancies. To improve performance on diverse audio recordings and scaling training, audio captioning and Audio-Language literature has used WavCaps Mei et al. (2024b) dataset. The WavCaps dataset is a large-scale, weakly-labelled audio captioning dataset designed for audio-language multimodal research. It comprises approximately 400k audio clips with paired captions. The audio clips are sourced from various web sources, including FreeSound, BBC Sound Effects, SoundBible, and the AudioSet Strongly-labelled Subset. The dataset uses ChatGPT to filter and transform raw descriptions into high-quality captions. As there is no human verification in the process, we only use this data for training and not for evaluation. To generate the audio difference explanation for WavCaps, we follow the same process listed in Section 2.1 and Section E. We call the resulting dataset Wav-Caps Difference (WCD) and its statistics are shown in Table 19. For example, in Tier 1, the WCD

| Task | Tier | BLEU$_1$ | BLEU$_2$ | BLEU$_3$ | BLEU$_4$ | METEOR | ROUGE$_L$ | CIDEr | SPICE | SPIDEr | AVG |
|------|------|----------|----------|----------|----------|--------|-----------|-------|-------|--------|-----|
| ACD | 1 | 0.303 | 0.199 | 0.130 | 0.082 | 0.184 | 0.319 | 0.175 | 0.082 | 0.128 | 0.178 |
| ACD | 2 | 0.396 | 0.263 | 0.188 | 0.135 | 0.176 | 0.349 | 0.156 | 0.152 | 0.154 | 0.219 |
| ACD | 3 | 0.439 | 0.271 | 0.192 | 0.149 | 0.175 | 0.319 | 0.0736 | 0.140 | 0.107 | 0.207 |
| CLD | 1 | 0.350 | 0.257 | 0.195 | 0.138 | 0.223 | 0.409 | 0.116 | 0.138 | 0.127 | 0.217 |
| CLD | 2 | 0.507 | 0.343 | 0.258 | 0.199 | 0.236 | 0.459 | 0.922 | 0.256 | 0.589 | 0.419 |
| CLD | 3 | 0.411 | 0.249 | 0.170 | 0.123 | 0.166 | 0.300 | 0.111 | 0.143 | 0.127 | 0.200 |
| ACD | 1 | 0.351 | 0.247 | 0.174 | 0.118 | 0.210 | 0.368 | 0.329 | 0.111 | 0.220 | 0.236 |
| ACD | 2 | 0.425 | 0.298 | 0.221 | 0.163 | 0.193 | 0.375 | 0.284 | 0.166 | 0.225 | 0.261 |
| ACD | 3 | 0.442 | 0.282 | 0.199 | 0.153 | 0.188 | 0.318 | 0.117 | 0.129 | 0.123 | 0.217 |
| CLD | 1 | 0.368 | 0.266 | 0.192 | 0.128 | 0.237 | 0.431 | 0.270 | 0.156 | 0.212 | 0.251 |
| CLD | 2 | 0.517 | 0.376 | 0.294 | 0.233 | 0.234 | 0.476 | 0.990 | 0.292 | 0.641 | 0.450 |
| CLD | 3 | 0.448 | 0.298 | 0.211 | 0.157 | 0.199 | 0.327 | 0.127 | 0.205 | 0.166 | 0.238 |
| ACD | 1 | 0.376 | 0.258 | 0.183 | 0.131 | 0.214 | 0.364 | 0.445 | 0.129 | 0.287 | 0.265 |
| ACD | 2 | 0.427 | 0.287 | 0.209 | 0.155 | 0.197 | 0.373 | 0.305 | 0.181 | 0.243 | 0.264 |
| ACD | 3 | 0.471 | 0.293 | 0.200 | 0.149 | 0.203 | 0.316 | 0.161 | 0.147 | 0.154 | 0.233 |
| CLD | 1 | 0.395 | 0.296 | 0.224 | 0.156 | 0.257 | 0.431 | 0.214 | 0.176 | 0.195 | 0.260 |
| CLD | 2 | 0.585 | 0.427 | 0.340 | 0.280 | 0.267 | 0.525 | 1.485 | 0.322 | 0.904 | 0.571 |
| CLD | 3 | 0.444 | 0.266 | 0.177 | 0.127 | 0.187 | 0.305 | 0.221 | 0.171 | 0.196 | 0.233 |

Table 16: Architecture results. The top half of the table uses baseline architecture with random audio encoder weights. The middle half of the table uses pretrained audio encoder weights. The bottom half of the table is ADIFF which modifies baseline architecture with separator token and cross-projection

| LM | Task | BLEU$_1$ | BLEU$_2$ | BLEU$_3$ | BLEU$_4$ | METEOR | ROUGE$_L$ | CIDEr | SPICE | SPIDEr | AVG |
|----|------|----------|----------|----------|----------|--------|-----------|-------|-------|--------|-----|
| Base | ACD-1 | 0.376 | 0.258 | 0.183 | 0.131 | 0.214 | 0.364 | 0.445 | 0.129 | 0.287 | 0.265 |
| Base | ACD-2 | 0.427 | 0.287 | 0.209 | 0.155 | 0.197 | 0.373 | 0.305 | 0.181 | 0.243 | 0.264 |
| Base | ACD-3 | 0.471 | 0.293 | 0.200 | 0.149 | 0.203 | 0.316 | 0.161 | 0.147 | 0.154 | 0.233 |
| Base | CLD-1 | 0.395 | 0.296 | 0.224 | 0.156 | 0.257 | 0.431 | 0.214 | 0.176 | 0.195 | 0.260 |
| Base | CLD-2 | 0.585 | 0.427 | 0.340 | 0.280 | 0.267 | 0.525 | 1.485 | 0.322 | 0.904 | 0.571 |
| Base | CLD-3 | 0.444 | 0.266 | 0.177 | 0.127 | 0.187 | 0.305 | 0.221 | 0.171 | 0.196 | 0.233 |
| Med. | ACD-1 | 0.401 | 0.264 | 0.178 | 0.119 | 0.211 | 0.373 | 0.520 | 0.131 | 0.325 | 0.280 |
| Med. | ACD-2 | 0.427 | 0.287 | 0.210 | 0.154 | 0.198 | 0.370 | 0.324 | 0.171 | 0.248 | 0.266 |
| Med. | ACD-3 | 0.468 | 0.300 | 0.210 | 0.160 | 0.201 | 0.337 | 0.304 | 0.160 | 0.232 | 0.264 |
| Med. | CLD-1 | 0.409 | 0.287 | 0.209 | 0.149 | 0.270 | 0.459 | 0.486 | 0.147 | 0.317 | 0.304 |
| Med. | CLD-2 | 0.560 | 0.410 | 0.324 | 0.264 | 0.287 | 0.510 | 0.941 | 0.340 | 0.641 | 0.475 |
| Med. | CLD-3 | 0.474 | 0.296 | 0.200 | 0.143 | 0.203 | 0.326 | 0.262 | 0.146 | 0.204 | 0.250 |
| Large | ACD-1 | 0.362 | 0.250 | 0.175 | 0.124 | 0.221 | 0.371 | 0.356 | 0.124 | 0.240 | 0.247 |
| Large | ACD-2 | 0.428 | 0.284 | 0.208 | 0.156 | 0.197 | 0.353 | 0.153 | 0.141 | 0.147 | 0.230 |
| Large | ACD-3 | 0.479 | 0.311 | 0.216 | 0.162 | 0.207 | 0.328 | 0.182 | 0.149 | 0.165 | 0.244 |
| Large | CLD-1 | 0.365 | 0.269 | 0.198 | 0.133 | 0.238 | 0.419 | 0.405 | 0.121 | 0.263 | 0.268 |
| Large | CLD-2 | 0.502 | 0.329 | 0.239 | 0.180 | 0.223 | 0.457 | 0.930 | 0.219 | 0.574 | 0.406 |
| Large | CLD-3 | 0.483 | 0.318 | 0.230 | 0.177 | 0.191 | 0.331 | 0.263 | 0.192 | 0.227 | 0.268 |
| XL | ACD-1 | 0.373 | 0.259 | 0.181 | 0.129 | 0.216 | 0.376 | 0.391 | 0.153 | 0.272 | 0.261 |
| XL | ACD-2 | 0.431 | 0.297 | 0.219 | 0.164 | 0.202 | 0.373 | 0.256 | 0.173 | 0.215 | 0.259 |
| XL | ACD-3 | 0.498 | 0.310 | 0.209 | 0.154 | 0.208 | 0.338 | 0.072 | 0.141 | 0.107 | 0.226 |
| XL | CLD-1 | 0.373 | 0.271 | 0.195 | 0.112 | 0.256 | 0.420 | 0.402 | 0.149 | 0.275 | 0.273 |
| XL | CLD-2 | 0.546 | 0.386 | 0.296 | 0.234 | 0.255 | 0.486 | 0.788 | 0.266 | 0.527 | 0.420 |
| XL | CLD-3 | 0.490 | 0.300 | 0.205 | 0.146 | 0.208 | 0.344 | 0.243 | 0.180 | 0.212 | 0.259 |

Table 17: Scaling language model results. The language model in ADIFF architecture (transformer decoder) ranges in size from 128 million (Base) to 1.5 billion parameters (XL).

Train split has 366k examples with explanations having a median of 27, a maximum of 58, and a vocabulary size of 38425, while Tier 3 has 366k examples and the explanations have a median of 155, a maximum of 234, and a vocabulary size of 51028.

We trained the ADIFF model using a three-stage training process on the WavCaps, ACD, and CLD datasets, and evaluated it on the ACD and CLD test sets. A summary of the results is shown in Table 21, with detailed metrics in Table 20. Although incorporating WCD into the training data improves performance on some metrics, it generally results in a performance decline on the ACD and CLD test sets. This performance drop after adding WavCaps has been observed in existing literature Mei et al. (2024b); Deshmukh et al. (2023a; 2024b), primarily because adding WavCaps changes the training distribution, shifting it further away from the testing distribution of ACD and CLD. This shift highlights a limitation of objective metrics that focus on linguistics. However, integrating WCD

| Task | Tier | BLEU$_1$ | BLEU$_2$ | BLEU$_3$ | BLEU$_4$ | METEOR | ROUGE$_L$ | CIDEr | SPICE | SPIDEr | AVG |
|------|------|----------|----------|----------|----------|--------|-----------|-------|-------|--------|-----|
| ACD | 1 | 0.384 | 0.256 | 0.182 | 0.129 | 0.208 | 0.352 | 0.469 | 0.130 | 0.300 | 0.268 |
| ACD | 2 | 0.444 | 0.304 | 0.223 | 0.162 | 0.195 | 0.377 | 0.407 | 0.185 | 0.296 | 0.288 |
| ACD | 3 | 0.477 | 0.305 | 0.214 | 0.164 | 0.199 | 0.323 | 0.148 | 0.152 | 0.150 | 0.237 |
| CLD | 1 | 0.446 | 0.338 | 0.258 | 0.191 | 0.304 | 0.478 | 0.832 | 0.190 | 0.511 | 0.394 |
| CLD | 2 | 0.520 | 0.380 | 0.299 | 0.236 | 0.242 | 0.489 | 1.108 | 0.270 | 0.689 | 0.470 |
| CLD | 3 | 0.439 | 0.277 | 0.190 | 0.136 | 0.188 | 0.340 | 0.069 | 0.179 | 0.124 | 0.216 |
| ACD | 1 | 0.405 | 0.275 | 0.192 | 0.135 | 0.221 | 0.381 | 0.469 | 0.137 | 0.303 | 0.280 |
| ACD | 2 | 0.440 | 0.301 | 0.232 | 0.180 | 0.197 | 0.379 | 0.518 | 0.173 | 0.345 | 0.307 |
| ACD | 3 | 0.485 | 0.317 | 0.225 | 0.171 | 0.208 | 0.331 | 0.204 | 0.162 | 0.183 | 0.254 |
| CLD | 1 | 0.433 | 0.338 | 0.265 | 0.203 | 0.302 | 0.488 | 1.093 | 0.211 | 0.652 | 0.443 |
| CLD | 2 | 0.519 | 0.362 | 0.274 | 0.213 | 0.235 | 0.464 | 1.124 | 0.260 | 0.692 | 0.460 |
| CLD | 3 | 0.522 | 0.355 | 0.255 | 0.191 | 0.220 | 0.360 | 0.634 | 0.200 | 0.417 | 0.350 |

Table 18: Audio grounding and finetuning results. The top half of the table shows ADIFF performance with position captioning. The bottom half of Table finetunes the language model of ADIFF.

| Data | Split | Examples per tier | Tier 1 | | | Tier 2 | | | Tier 3 | | |
|------|-------|-------------------|--------|-----|-------|--------|-----|-------|--------|-----|-------|
| | | | Med. | Max | Vocab. | Med. | Max | Vocab. | Med. | Max | Vocab. |
| CLD | Train | 19195 | 27 | 49 | 6528 | 51 | 92 | 6462 | 155 | 221 | 10818 |
| CLD | Val | 5225 | 27 | 46 | 3743 | 51 | 89 | 4024 | 154 | 223 | 7026 |
| CLD | Test | 5225 | 27 | 28 | 5225 | 52 | 86 | 4059 | 156 | 219 | 7152 |
| ACD | Train | 48660 | 29 | 47 | 3287 | 52 | 104 | 8483 | 155 | 235 | 12891 |
| ACD | Val | 2456 | 28 | 46 | 2350 | 52 | 87 | 2563 | 154 | 227 | 4566 |
| ACD | Test | 4680 | 29 | 47 | 3287 | 53 | 95 | 3329 | 154 | 220 | 5489 |
| WCD | Train | 366083 | 28 | 58 | 38425 | 53 | 101 | 38277 | 155 | 234 | 51028 |

Table 19: Comparing WavCaps Difference (WCD) dataset statistics with AudioCaps Difference (ACD) and Clotho Difference (CLD) dataset

into ADIFF significantly enhances the model's vocabulary at inference, increases the coverage of sounds and audio sources, and provides greater granularity compared to ADIFF without WCD.

| Task | Models | Tier 1 | | | Tier 2 | | | Tier 3 | | |
|------|--------|--------|--------|--------|--------|--------|--------|--------|--------|--------|
| | | BLEU$_4$ | METEOR | SPIDEr | BLEU$_4$ | METEOR | SPIDEr | BLEU$_4$ | METEOR | SPIDEr |
| ACD | ADIFF | **0.135** | **0.221** | **0.303** | 0.180 | 0.197 | **0.345** | **0.171** | **0.208** | **0.183** |
| | ADIFF + WCD | 0.105 | 0.189 | 0.198 | **0.197** | **0.205** | 0.259 | 0.152 | 0.197 | 0.152 |
| CLD | ADIFF | **0.203** | **0.302** | **0.652** | 0.213 | 0.235 | 0.692 | **0.191** | **0.220** | **0.417** |
| | ADIFF + WCD | 0.126 | 0.246 | 0.278 | **0.322** | **0.290** | **1.022** | 0.180 | 0.211 | 0.367 |

Table 20: ADIFF performance with WavCaps difference explanation (WCD) added to training data. The top half of the table shows the performance of models on the task of ACD, while The bottom half of the table shows the performance of models on the task of CLD. The results with all metrics and the average score are available in Appendix Table 21.

| Task | Tier | BLEU$_1$ | BLEU$_2$ | BLEU$_3$ | BLEU$_4$ | METEOR | ROUGE$_L$ | CIDEr | SPICE | SPIDEr | AVG |
|------|------|----------|----------|----------|----------|--------|-----------|-------|-------|--------|-----|
| ACD | 1 | 0.405 | 0.276 | 0.192 | 0.135 | 0.221 | 0.381 | 0.469 | 0.137 | 0.303 | 0.280 |
| ACD | 2 | 0.440 | 0.301 | 0.232 | 0.180 | 0.197 | 0.379 | 0.518 | 0.173 | 0.345 | 0.307 |
| ACD | 3 | 0.486 | 0.317 | 0.225 | 0.171 | 0.208 | 0.331 | 0.204 | 0.162 | 0.183 | 0.254 |
| CLD | 1 | 0.433 | 0.338 | 0.265 | 0.203 | 0.302 | 0.488 | 1.093 | 0.211 | 0.652 | 0.443 |
| CLD | 2 | 0.519 | 0.362 | 0.274 | 0.213 | 0.235 | 0.464 | 1.124 | 0.260 | 0.692 | 0.460 |
| CLD | 3 | 0.522 | 0.355 | 0.255 | 0.191 | 0.220 | 0.360 | 0.634 | 0.200 | 0.417 | 0.350 |
| ACD | 1 | 0.346 | 0.226 | 0.155 | 0.105 | 0.189 | 0.323 | 0.292 | 0.104 | 0.198 | 0.215 |
| ACD | 2 | 0.472 | 0.342 | 0.257 | 0.197 | 0.205 | 0.399 | 0.338 | 0.179 | 0.259 | 0.294 |
| ACD | 3 | 0.463 | 0.284 | 0.198 | 0.152 | 0.197 | 0.304 | 0.161 | 0.142 | 0.152 | 0.228 |
| CLD | 1 | 0.368 | 0.269 | 0.196 | 0.126 | 0.246 | 0.401 | 0.461 | 0.096 | 0.278 | 0.271 |
| CLD | 2 | 0.599 | 0.467 | 0.387 | 0.322 | 0.290 | 0.546 | 1.73 | 0.318 | 1.022 | 0.631 |
| CLD | 3 | 0.505 | 0.328 | 0.235 | 0.180 | 0.211 | 0.348 | 0.564 | 0.170 | 0.367 | 0.323 |

Table 21: ADIFF performance with WavCaps difference explanation (WCD) added to training data. The top half of the table shows the ADIFF model performance. The bottom part shows the performance of ADIFF with WCD added to the training data. The Tier 1, 2, and 3 classifications correspond to the explanation tiers detailed in Section 2.

---

**LLM prompt**

---

You are a helpful assistant with expert knowledge about audio, acoustics, and psychoacoustics. You study audio, which is the study of sound and its properties. You study acoustics, which revolve around the generation, propagation, and reception of sound waves. You study Psychology which posits that a sound is a complex stimulus that encompasses a vast range of acoustic properties involving aspects of cognition, psychoacoustics, and psychomechanics. Your task is to rate the difference explanation between the two audios and provide a score for how detailed and granular is the audio difference explanation. Pay attention to differences in terms of audio events (dog barking vs cat meowing), sound sources (musical instrument, human voice), acoustic scene (park vs living room), signal characteristics (frequency, pitch, loudness, etc) emotion response (listening to the audio invokes a sense of happiness, fear, in you) and any other difference using world-knowledge or using deductive reasoning. The score can take the value of 1,2,3,4, and 5. The score of 1 (poor) indicates the explanation is very vague, lacking detail and specificity. The score of 2 (fair) indicates the explanation provides some detail but is generally too broad. The score of 3 (good) indicates the explanation is moderately detailed, covering key aspects but missing finer points. The score of 4 (Very Good) indicates the explanation is detailed, covering most aspects with good specificity. 5 (Excellent) the explanation is highly detailed, thoroughly covering all relevant aspects with precise specificity. Return your rating as a JSON with the key "score".

---

Table 22: Prompting LLM to score information density in the generated audio difference explanation

## Q   INFORMATION DENSITY IN DIFFERENCE EXPLANATION

We create the three Tiers in Section 2 with each tier containing more detailed and granular information than the previous Tier. We explain the data statistics in 2.1 and further analysis on difference explanations in F. In this section, we demonstrate the quality of the three tiers, showing that Tier-3 offers greater granularity in descriptions compared to Tier-1. We propose three ways to verify and prove this.

### Q.1   QUALITATIVE ANALYSIS AND INFORMATION DENSITY SCORE

Qualitatively, one can review the data files and compare them. Each entry contains audio filepath1, audio filepath2, and an answer. This format is consistent across the data, making the difference explanations directly comparable across the ACD test set Tier-1, Tier-2, and Tier-3. Manual inspection will reveal that Tier-3 contains more information or is denser than Tier-2 compared to Tier-1.

Quantitatively, we can construct metrics that measure the density of information per difference explanation and compare these metrics across tiers. One proxy metric is the unique vocabulary used across the three tiers, shown in data statistics Table 1. We can see that the unique vocabulary in Tier-3 is higher than in Tier-2 and Tier-1, indicating more vocabulary and concepts introduced in Tier-3. However, more vocabulary does not necessarily imply more information, especially concerning audio characteristics. Therefore, we propose an information density score, asking human annotators to rate each explanation between 1 and 5. Due to budget constraints, we use an LLM to simulate human evaluation and provide scores between 1 and 5. The LLM prompt used is shown in Table 22. We use this prompt to get scores for each difference explanation across the ACD and CLD datasets. Then we average the score across each Tier of difference explanation. The results are shown in Table 23.

|  | Tier-1 | Tier-2 | Tier-3 |
|---|---|---|---|
| Avg. score | 2.91 | 3.64 | 4.17 |

Table 23: Information density score across the three Tiers. The Avg. density score is obtained by averaging the density score per difference explanation averaged across the ACD and CLD explanations for each Tier.

The results demonstrate that Tier-3 information density scores are the highest, while Tier-1 scores are the lowest. Although this metric serves as a proxy for human evaluation, the difference of at least 0.5 between the scores, averaged over 85k examples per tier (ACD + CLD), indicates that Tier-3 explanations offer greater granularity and information in descriptions compared to Tier-1 explanations.

### Q.2   ENTROPY OF EACH TIER

Each Tier of the ACD and CLD dataset contains different levels of information. To measure this information one can use entropy. Entropy is a measure of uncertainty or randomness in a dataset,

and computing it for a corpus at the character and word levels will provide us insights into the structure and predictability of the language. Character-level entropy quantifies the variability in how individual characters are used and combined, reflecting language complexity, encoding efficiency, and text predictability. Word-level entropy, on the other hand, captures the diversity and distribution of words, shedding light on lexical richness and the underlying linguistic patterns. To compute entropy, we use the Shannon's formula:

$$H = -\sum p(x) \log_2 p(x) \tag{4}$$

where $p(x)$ is the probability of a character or word occurring in the corpus. One note is we assume a simple unigram model, where each token's probability is independent of others. For character-level entropy, we count the frequency of each character. Normalize these frequencies to convert to valid probabilities and compute entropy by equation 4. For word-level entropy, we tokenize the corpus into words, compute their frequencies, and normalize as before to obtain probabilities. Finally, we sum the entropy contributions for each unique word and compute entropy by equation 4.

| Entropy | Tier-1 | Tier-2 | Tier-3 |
|---|---|---|---|
| Character | 4.28 | 4.35 | 4.54 |
| Word | 7.89 | 8.01 | 8.50 |

Table 24: Character and word entropy across the three Tiers. For the entropy calculation, we construct the corpus for each Tier. This involves combining ACD and CLD data for each Tier across the three splits of dev, val, and test.

The results are shown in Table 24. A higher entropy at the character or word level suggests greater diversity or unpredictability in the corpus. A higher character entropy would mean the text has a wide variety of characters with a relatively even distribution (e.g., many unique letters, numbers, punctuation marks, etc). That is the text is less redundant, meaning characters appear less predictably. Similarly, a higher word entropy would mean the corpus has a rich vocabulary with words appearing in a more evenly distributed manner. That is the text is less predictable at the word level, indicating diverse topics or language use. Across datasets, the character-level entropy is relatively stable, implying the character set and its usage (letters, punctuation, spaces, etc.) don't vary significantly between corpora. The word-level entropy is higher for Tier-3, indicating more complex and varied vocabulary. Lower entropy in Tier-1 suggests more uniform and constrained language, which is expected for shorter texts or structured responses. We also computed the numbers per data source and found the CLD datasets tend to have slightly higher word-level entropy compared to the ACD datasets (e.g., detail answers have ∼8.52 vs. ∼8.43) which reflect differences in diversity of text content as ACD is sourced from AudioSet which mainly contains speech. From a model development perspective, higher word-level entropy datasets (e.g., Tier-3) will be more challenging for models, requiring greater capacity to handle vocabulary diversity. Though entropy allows us to measure uncertainty in a dataset, it does not measure semantic information, which would be measured in the density score proposed in Section Q.1.

| Task | Tier | BLEU$_1$ | BLEU$_2$ | BLEU$_3$ | BLEU$_4$ | METEOR | ROUGE$_L$ | CIDEr | SPICE | SPIDEr | AVG |
|---|---|---|---|---|---|---|---|---|---|---|---|
| ACD | 1 | 0.405 | 0.276 | 0.192 | 0.135 | 0.221 | 0.381 | 0.469 | 0.137 | 0.303 | 0.280 |
| ACD | 2 | 0.440 | 0.301 | 0.232 | 0.180 | 0.197 | 0.379 | 0.518 | 0.173 | 0.345 | 0.307 |
| ACD | 3 | 0.486 | 0.317 | 0.225 | 0.171 | 0.208 | 0.331 | 0.204 | 0.162 | 0.183 | 0.254 |
| CLD | 1 | 0.433 | 0.338 | 0.265 | 0.203 | 0.302 | 0.488 | 1.093 | 0.211 | 0.652 | 0.443 |
| CLD | 2 | 0.519 | 0.362 | 0.274 | 0.213 | 0.235 | 0.464 | 1.124 | 0.260 | 0.692 | 0.460 |
| CLD | 3 | 0.522 | 0.355 | 0.255 | 0.191 | 0.220 | 0.360 | 0.634 | 0.200 | 0.417 | 0.350 |
| ACD | 1 | 0.411 | 0.287 | 0.207 | 0.148 | 0.217 | 0.388 | 0.220 | 0.127 | 0.174 | 0.242 |
| ACD | 2 | 0.497 | 0.365 | 0.289 | 0.234 | 0.234 | 0.437 | 0.200 | 0.222 | 0.211 | 0.299 |
| ACD | 3 | 0.500 | 0.321 | 0.226 | 0.169 | 0.211 | 0.340 | 0.117 | 0.171 | 0.144 | 0.244 |
| CLD | 1 | 0.412 | 0.295 | 0.215 | 0.152 | 0.219 | 0.402 | 0.163 | 0.119 | 0.141 | 0.235 |
| CLD | 2 | 0.491 | 0.368 | 0.296 | 0.245 | 0.230 | 0.445 | 0.156 | 0.231 | 0.194 | 0.295 |
| CLD | 3 | 0.495 | 0.320 | 0.227 | 0.171 | 0.203 | 0.337 | 0.088 | 0.167 | 0.128 | 0.237 |

Table 25: ADIFF performance with top-k and top decoding vs greedy decoding. The top half of the table shows the ADIFF model performance with top-k and top-p decoding. The bottom part shows the performance of ADIFF model with greedy decoding. We observe overall improvement in performance with top-p and top-k decoding. The Tier 1, 2, and 3 classifications correspond to the explanation tiers detailed in Section 2.

