# OpenReview forum: "ADIFF: Explaining audio difference using natural language"
_ICLR.cc/2025/Conference — ICLR 2025 Spotlight_

### Official Review · Reviewer_6ejz · 2024-11-02

**Soundness:** 3
**Presentation:** 3
**Contribution:** 3
**Rating:** 6
**Confidence:** 4

**Summary:**

The authors propose a new derived dataset from AudioCaps and Clotho for describing differences between audios in three different tiers corresponding to the complexities of outputted descriptions. They also propose ADIFF, an architecture improving upon baseline by adding a separator token and cross-projection layer for various captioning metrics.

**Strengths:**

- The proposed task in audio difference is novel and interesting. This is a good contribution to the audio reasoning research community. The three-tier approach makes sense and these generated descriptions can be utilized in various research topics.
- The authors provide thorough ablation studies in proposed architectures including architectural choices, language model sizes, and training scenarios, etc.

**Weaknesses:**

- The choices of captioning metrics seem insufficient. As also observed by the authors in section 5.1, these metrics can be biased to the words irrelevant to the differences in audio. To complement the metrics, it might worth adding other benchmark such that the differences in sounding objects can be captured from the SED output from each audio, or the characteristics of audio can be derived from other signal processing techniques?
- Showing the complete captioning metrics in all the tables (2, 4, 5, 6) does not seem to help in the narratives, especially not many of them are referred in the text. It might worth consolidate and only show a couple, for example, in the audio captioning research, the community is starting to converge to several more representative metrics that are more relevant to the task 1. Also it might be helpful to bulk the high performant numbers that are also referred to in the text.
- In table 1, is the number of examples from the number of audio in each dataset? It seems that the proposed task requires two audio, therefore the dataset should be consisted of {a_i, a_j, e_i}? If this is the case, how many pairs are included in each of the split? Also, in the original dataset, each audio can include more than 1 caption (except the train split in AudioCaps), how are these captions selected as the seed for LLMs to generate descriptions, these are not clearly explained in the main paper nor in the appendix.

1 https://dcase.community/challenge2024/task-automated-audio-captioning#evaluation

**Questions:**

- What are the types of differences described in different tiers? Is there a distribution around differences in sounding objects, differences in sounding characteristics within the same category of sound? It would be really helpful to provide these analysis from LLM generated captions.
- For table 3, is the human evaluation also based on the difference between two audios? If so, other than the three datasets utilized (Studio, FSD50K, GTZAN), is there a reason why proposed ACD and CLD are not included in the human evaluation?

---

> ### Author Response · Authors · 2024-11-14
> **Rebuttal by Authors**
>
> We thank the reviewer for recognizing our contributions and providing constructive feedback. We address every question and hope that our response resolves your concerns. Any follow-up questions are welcome.
>
> ---
> **The choices of captioning metrics seem insufficient ... other signal processing techniques?**\
> The reviewer brings up a valid point on the limitation of metrics. We make similar observations in Section 5.1. Comparing to the link pointed by reviewer, we have all the metrics listed there except the SPIDER-FL and FENSE metric. However, even the SPIDEr-FL and FENSE metric fails to take into account the audio and signal information that the reviewer suggests. Therefore, to overcome the limitation of objective metrics, we decided to conduct human evaluations whose setup is explained in Appendix H.2 and results in Section 4 and Table 3. For the ablation studies, we rely on objective metrics, as human evaluation for each ablation study would be too costly given the size of the test set.
>
> **Showing the complete captioning metrics in all the tables (2, 4, 5, 6) ... referred to in the text.**\
> We thank the reviewer for this suggestion. We address this in the global comment under question 2. We hope this improves the readability of the numbers and clarity of the paper.
>
> **In table 1, is the number of examples from the number of audio in each dataset? ... these are not clearly explained in the main paper nor in the appendix.**\
> We thank the reviewer for bringing this up. The reviewer is right and the dataset indeed consists of {a_i, a_j, e_i}. We have updated this text in the paper and apologize for any confusion caused. For dataset statistics, the Table 1 is correct and indicates the triplets {a_i, a_j, e_i} in each dataset and split respectively. For example, the Train set of CLD consists of 19195 triplets for each Tier. For generating data, we perform random sampling where the sampling is done per audio, this removes the edge case of multiple captions. This is done in three steps. First, we flattent the .csvs to be audio file and single caption. For Clotho which has 5 captions, this results in 5 entries for each audio. Second, we perform sampling. For example, let's say the dataset contains N audios. We will loop through the N audios, for any ith audio we will randomly pick the second audio (jth) not including i to i+4th audio. This is implemented as a constructing index list excluding i, i+1,..i+4 index and sampling from this index list. Third, the selected ith and jth audio are then used to look up the caption for the ith and randomly picked jth audio. This caption then becomes the seed for LLM to generation difference e_i. We added this description on sampling in Appendix E. We again apologize about the text error and hope this clarifies the question!
>
> **What are the types of differences described in different tiers? ... analysis from LLM generated captions.**\
> We appreciate the reviewer's question and understand the importance of exploring the LLM-generated captions across the three Tiers. We can perform the analysis, by splitting text into words, converting them to lowercase, and focusing on words related to audio, we can identify words that appear in Tier 2 but not in Tier 1, showing unique audio characteristics used for the difference explanation in Tier 2. Similarly, we can use the same method to examine the distinctions between Tier 2 and Tier 3, where the differences often lie in more detailed and immersive sound characteristic descriptions. The results of this analysis are shown in Figure 7 with the description in Section F. Overall, we see the audio characteristics used in Tier 3 over Tier 2 (Figure 7, right subplot) and Tier 2 over Tier 1 (Figure 7, left subplot) are different, and focus on the different aspect and details between two audios.
>
> **For table 3, is the human evaluation also based on the difference between two ... not included in the human evaluation?**\
> The reviewer brings up a valid question. We specifically choose to source audio files from different datasets instead of ACD and CLD for human evaluation for two reasons. First, we wanted to compare against other Audio-Language Models like Qwen-Audio which supports two audios. By training naive baseline and ADIFF on the train set of ACD and CLD, we will be inherently matching the distribution of the test dataset both in terms of audio and language used. This is unfair to Qwen-Audio which might not have seen similar type of audio recordings. Therefore, to have a fair comparison, we didn't utilize audios from ACD and CLD test set in the human evaluation. Second, we wanted to see how performance changes across difference types of audios. For example, different acoustic scenes, same audio events, different domains, etc. By sourcing audios from Studio, FSD50K, GTZAN we were able to get the desired level of control and simulate the scenarios explained in Section 4. We hope the above answer helps.

---

> > ### Author Response · Authors · 2024-11-23
> > **Request to review the rebuttal**
> >
> > Thank you reviewer 6ejz for reviewing our paper. We have addressed your questions in our rebuttal response and uploaded a revised version of the paper. As the rebuttal period is nearing its conclusion, we kindly request you to review our rebuttal and share any additional comments or concerns you may have. Thank you once again for your valuable feedback!

---

> > > ### Comment · Reviewer_6ejz · 2024-11-25
> > >
> > > Thanks to the authors for your thorough responses. Most of the concerns has been addressed and already integrated in the manuscript. I will therefore increase my rating.

---

### Official Review · Reviewer_syVc · 2024-11-03

**Soundness:** 4
**Presentation:** 3
**Contribution:** 2
**Rating:** 8
**Confidence:** 4

**Summary:**

The authors introduces a new task, to explain differences between audio recordings using natural language. The authors show that current audio language models cannot complete this task, and propose AudioCaps and Clotho "Difference" sets, generated by LLM and verified by human in test set. These sets are trained using ADIFF, the proposed model, to bring performance boost.

**Strengths:**

The paper proposes a valid methodology in generating "audio difference" captions through LLMs using human captions; the methodology is verified through experiments.

**Weaknesses:**

1. The three-tier captioning process seems arbitary; no ablation study is conducted for this (i.e. if trained only on tier-3, can the models have good performance on tier-1? If they are purely hierarchical, then training on tier-3 should in theory yield good results on lower tiers.)
2. Both AudioCaps and Clotho's captioning quality is questionable; WavCaps and other larger caption datasets contain more diversity, yet no experiment is conducted on them.
3. The SOTA model, Qwen-Audio, seems to not be finetuned for this task as well. It prompts the question that finetuning a stronger model with out ADIFF may solve the problem entirely just through the proposed dataset.

**Questions:**

I suggest the authors complete the ablation experiments and additional finetuning experiments suggested.

---

> ### Author Response · Authors · 2024-11-14
> **Rebuttal by Authors**
>
> We thank the reviewer for recognizing our contributions. We want to highlight that our main contribution is the creation of the two difference datasets which are human-verified, then the second contribution is the baseline and the proposed ADIFF model, and lastly the various presented findings. We address every concern point by point and hope that our response resolves your concerns.
>
> ---
> **The three-tier captioning process seems arbitary; no ablation study is conducted for this (i.e. if trained only on tier-3, can the models have good performance on tier-1? If they are purely hierarchical, then training on tier-3 should in theory yield good results on lower tiers.)** \
> The three-tier captioning process is designed for evaluation purposes and to benchmark the comparative reasoning abilities of Audio-Language Models. The motivation for creating a three-tier dataset is detailed in Section 2. This motivation is rooted in auditory psychoacoustics, particularly sound perception studies, which examine how humans perceive sound, including how the body receives sound and how the brain interprets it. Human explanations draw from a diverse array of information sources, including acoustic details, human perception, and linguistic nuances. This diversity allows us to classify audio difference explanations into three tiers, with each tier providing more information than the previous one. Tier-3 explanations offer greater granularity in descriptions compared to Tier-1 explanations; however, this does not imply that Tier-1 descriptions are not as good as Tier-3 descriptions.
>
> **Both AudioCaps and Clotho's captioning quality is questionable;**\
> AudioCaps and Clotho are the only two human-annotated datasets available in audio-language literature. Therefore, we use their audio and captions as the source for creating our dataset. Additionally, we employ human annotators to verify the descriptions in our test set, ensuring the removal of any questionable captions.
>
> **WavCaps and other larger caption datasets contain more diversity, yet no experiment is conducted on them.**\
> We appreciate the reviewer’s suggestion. In current audio-language literature, AudioCaps and Clotho are used for evaluation. WavCaps, being LLM-generated and lacking human supervision, has never been used for evaluation as its correctness cannot be guaranteed. Additionally, AudioCaps and Clotho are considered diverse enough for evaluation purposes. However, as an alternative, it makes sense to use WavCaps to generate additional training data for differences and then test on the human-verified ACD and CLD test sets. We added this experiment, and the results are presented in Section P, Tables 20 and 21. We observe that adding WavCaps improves vocabulary and coverage of sound sources in produced audio difference explanations, though it might not be directly reflected in the linguistic metrics. The analysis is available in Section P and we will release both the newly created WCD dataset and the model checkpoint. We hope this addresses the reviewer's question.
>
> **The SOTA model, Qwen-Audio, seems to not be finetuned for this task as well. It prompts the question that finetuning a stronger model with out ADIFF may solve the problem entirely just through the proposed dataset.**\
> We understand the reviewer's point and address this in the global comment question 1. We have added two more baselines in the objective and subjective evaluation section: First, Qwen-AC with LoRA fine-tuned on the training set of ACD and CLD, and second, Qwen-AC with full fine-tuning (no frozen parameters) on the training set of ACD and CLD. The results are shown in Table 2 and Table 3, with analysis in Section 4.2. We hope this addresses any concerns on the baseline.

---

> > ### Comment · Reviewer_syVc · 2024-11-14
> > **Thanks for author's efforts!**
> >
> > Thanks the authors for their added baselines, analysis on WavCaps!
> >
> > "This motivation is rooted in auditory psychoacoustics, particularly sound perception studies, which examine how humans perceive sound, including how the body receives sound and how the brain interprets it." Any literature that specifically points to this? I feel like this is questionable, especially the claim that "Tier-3 explanations offer greater granularity in descriptions compared to Tier-1 explanations; however, this does not imply that Tier-1 descriptions are not as good as Tier-3 descriptions." In this case, how is the quality of these tiers measured? I imagined it has something to do with how "dense" or how much information is available in these captions.
> >
> > I will raise my rating to 7 with a good justification of the three-tier system.

---

> > > ### Author Response · Authors · 2024-11-15
> > > **Rebuttal by Authors**
> > >
> > > We sincerely thank the reviewer for reading our updated sections and rebuttal. For any additional clarifications, we are happy to address them.
> > >
> > > ---
> > > We understand reviewer's request and try to convert it into actionable questions that we can test and answer. So far we understand reviewer wants us to (1) "This motivation is rooted in auditory psychoacoustics, particularly sound perception studies, which examine how humans perceive sound, including how the body receives sound and how the brain interprets it."  Any literature that specifically points to this? (2) demonstrate the quality of the three tiers, showing that Tier-3 offers greater granularity in descriptions compared to Tier-1. If we have misunderstood the request, please correct us.
> > >
> > > (1) Our motivation is drawn from Section 33.4 of the chapter [1] in Perspectives on Auditory Research and the article [2] published in Frontiers in Psychology, 2019. We want to specifically highlight the introduction in [2], which we quote here "Regardless of the specific kind of sound one hears (whether it is speech, music, or another environmental sound) this process appears to involve (1) a multidimensional set of acoustic cues, (2) a process of matching cues to representations of objects and events stored in memory, and (3) the use of those object representations to organize auditory scenes". The authors of [2] then empirically analyze the three steps with respect to speech, sounds, and music in the discussion section of [2]. For the difference explanation, there is an additional component of detail or density in the explanation. Therefore, we roughly translate the three tiers to the difference explanation in Section 2 as (1) a multidimensional set of acoustic cues --> "key characteristics of each audio without much elaboration" (2) matching cues to representations of objects and events --> "It not only describes the sounds but also includes audio events, sound sources, the nature of the sounds, making it slightly more analytical" (3) use of those object representations to organize auditory scenes --> "delves into the sonic qualities of each audio, including potential sources and the listener’s experience. It compares the two audios in terms of complexity and engagement, providing a richer and more immersive description. It analyzes the audio events, acoustic scenes, sound sources, signal characteristics, tonal differences, and overall feel of each audio." We hope this clarifies the reviewer's question!
> > >
> > > (2) We offer two ways to verify and prove this. We are also open to alternate experiment setups that the reviewer suggests or would like us to run:
> > > - **Qualitative Analysis.** Practically, one can review the .csv files provided in the supplementary material. Each entry in the .csv contains audio filepath1, audio filepath2, and an answer. This format is consistent across the .csvs, and the ith and jth entries for ACD and CLD use the same audio file path. This makes the difference explanations directly comparable across the ACD test set Tier-1, ACD test set Tier-2, and ACD test set Tier-3. Manual inspection will reveal that Tier-3 contains more information or is denser than Tier-2 compared to Tier-1.
> > > - **Quantitative Analysis.** We can construct metrics that measure the density of information per difference explanation and compare these metrics across tiers. One proxy objective metric is the unique vocabulary used across the three tiers, shown in the data statistics in Table 1. We can see that the unique vocabulary in Tier-3 is higher than in Tier-2 and Tier-1, indicating more vocabulary and concepts introduced in Tier-3. However, more vocabulary does not necessarily imply more information, especially concerning audio characteristics. Therefore, we need an information density score that we can assign to each explanation and compare across tiers. To achieve this, we can ask human annotators to provide an information density score between 1 and 5 and compare these scores across tiers. However, we do not have the annotation budget for such a large-scale experiment. Instead, we use an LLM to simulate human evaluation and provide an information density score between 1 and 5. The LLM prompt, results, and analysis are available in Table 22, Table 23, and Section Q, respectively. We see that the information density score for Tier-1 is 2.91, Tier-2 is 3.64, and Tier-3 is 4.17. Given these scores, we can state that Tier-3 explanations offer greater granularity and information compared to Tier-1 explanations.
> > >
> > > We hope the above answers the reviewer's question on the three-tier system.
> > >
> > > [1] Perspectives on Auditory Research published by Springer, https://link.springer.com/chapter/10.1007/978-1-4614-9102-6_33?form=MG0AV3#Sec8 \
> > > [2] "Acoustic Correlates of Auditory Object and Event Perception: Speakers, Musical Timbres, and Environmental Sounds", Frontiers in Psychology, 2019, https://www.frontiersin.org/journals/psychology/articles/10.3389/fpsyg.2019.01594/full?form=MG0AV3

---

> > > > ### Author Response · Authors · 2024-11-23
> > > > **Request to review the rebuttal**
> > > >
> > > > Thank you reviewer syVc for reviewing our paper. We have addressed your questions in our rebuttal response and uploaded a revised version of the paper. As the rebuttal period is nearing its conclusion, we kindly request you to review our rebuttal and share any additional comments or concerns you may have. Thank you once again for your valuable feedback!

---

> > > > > ### Comment · Reviewer_syVc · 2024-11-25
> > > > > **Update**
> > > > >
> > > > > Thank the authors for the comprehensive response! I suggest reframing the analysis from an information theory perspective. Each stage likely contains different levels of information entropy, requiring the captioning system to navigate through varying degrees of uncertainty to arrive at accurate answers. Analyzing the perplexity at different stages could provide valuable insights into how the system handles these varying entropy levels. Including such analysis would strengthen the paper's theoretical foundation (and provide empirical justification to the three-tier system) significantly.
> > > > >
> > > > > I raise my score to 7. Should the authors incorporate the suggested perplexity analysis, I would further raise the score to 8.
> > > > >
> > > > > Thank the authors for their great work. Best of luck!

---

> ### Author Response · Authors · 2024-11-26
> **Rebuttal by authors**
>
> We sincerely thank the reviewer for the suggestion and the help provided in improving the manuscript.
>
> ---
> We add an analysis from an information theory perspective to Appendix Section Q, specifically Appendix Section Q2 and Table 24. The analysis focuses on Entropy for each tier of audio difference explanation. The perplexity would simply be 2 raised to the entropy value.
>
> The results are shown in Table 24 and we provide a summary here. A higher entropy at the character or word level suggests greater diversity or unpredictability in the corpus. A higher character entropy would mean the text has a wide variety of characters with a relatively even distribution (e.g., many unique letters, numbers, punctuation marks, etc). Similarly, a higher word entropy would mean the corpus has a rich vocabulary with words appearing in a more evenly distributed manner. We see a higher word-level entropy for Tier 3 signifying a more complex and varied vocabulary, whereas lower entropy in Tier-1 indicates a more uniform and constrained language, typical of shorter texts or structured responses. We also computed the numbers per data source and found the CLD datasets tend to have slightly higher word-level entropy compared to the ACD datasets, which reflects differences in the diversity of text content as ACD is sourced from AudioSet which mainly contains speech. From a model development perspective, higher word-level entropy datasets (e.g., Tier-3) will be more challenging for models, requiring greater capacity to handle vocabulary diversity. The full explanation can be found in Appendix Section Q2.

---

> > ### Comment · Reviewer_syVc · 2024-11-26
> > **Response to Rebuttal**
> >
> > Thank the authors for the quick update! I raise my score to 8.

---

### Official Review · Reviewer_uxzd · 2024-11-04

**Soundness:** 4
**Presentation:** 3
**Contribution:** 4
**Rating:** 8
**Confidence:** 5

**Summary:**

This paper introduces a novel task focused on understanding and explaining the differences between audio recordings through various human perception primitives. It proposes a language model-based (LM-based) approach to generate explanations for the differences between two audio samples, establishing a pipeline that utilizes a large language model and existing audio metadata to create different tiers of audio differences. The experiments evaluate various configurations of the proposed audio difference model, considering factors such as text-encoder size, model components, and multi-stage training paradigms. Additionally, the paper discusses the challenges associated with creating annotations and explores potential real-world applications for this task.

**Strengths:**

The paper contains three strengths:

1.  The paper introduces a considerable novelty by defining a new audio task inspired by the development of multimodal language models. The approach to explaining audio differences effectively enhances the LLM's ability to understand audio inputs at both the semantic and acoustic levels. Additionally, the method incorporates primitives of human perception into the dataset creation process, progressively decomposing audio elements from audio events to acoustic scenes, signal-level analysis, emotions, and other properties. This enriches the data quality and practical usability.

2. The evaluation is thorough, primarily focusing on ablation studies. The experimental framework is well-structured, starting with a baseline model and a reference state-of-the-art model (Qwen-Audio Chat), followed by ablation studies that investigate different design choices within the proposed model.

3. The paper includes thoughtful discussions regarding the hallucination tendencies of the LLM when generating annotations. It provides insights into how the authors filter these data using existing methods to enhance dataset quality.

Overall, I find this paper valuable and believe it makes significant contributions to the audio community.

**Weaknesses:**

There is one weakness of this paper:

1. The presentation of the experimental section requires further refinement. The experimental tables are somewhat difficult to follow, as the model names are not consistently listed in most of them. Additionally, highlighting or bolding the best values for each metric would enhance clarity and emphasize the differences more effectively.

**Questions:**

1.Regarding the structure of the sequence sent to the cross-projection layer in Figure 3, what is the length of the audio prefixes (i.e., [prefix 1] and [prefix 2])? From my understanding, HTS-AT provides only one audio embedding for the entire segment of audio samples. Do [prefix 1] and [prefix 2] consist of just a single token representing the latent embedding from HTS-AT?

I suggest including a more detailed explanation of this aspect, rather than solely focusing on the right side of Figure 3.

---

> ### Author Response · Authors · 2024-11-14
> **Rebuttal by Authors**
>
> We sincerely thank the reviewer for recognizing our contribution and novelty! We hope that our response resolves your concerns. Any follow-up questions are welcome.
>
> ---
> **The presentation of the experimental section requires further refinement. The experimental tables are somewhat difficult to follow, as the model names are not consistently listed in most of them. Additionally, highlighting or bolding the best values for each metric would enhance clarity and emphasize the differences more effectively.**\
> We apologize for the difficulty in reading the experimental section. We address this in the global comment under question 2. We have updated the PDF and retained only the BLUE_4, METEOR, and SPIDEr metric in the main paper, while moving the full tables to the Appendix. Moreover, we have bolded the best-performing number for each metric in the Table. We hope this improves the readability of the numbers and clarity of the paper.
>
> **Regarding the structure of the sequence sent to the cross-projection layer in Figure 3, what is the length of the audio prefixes (i.e., [prefix 1] and [prefix 2])? From my understanding, HTS-AT provides only one audio embedding for the entire segment of audio samples. Do [prefix 1] and [prefix 2] consist of just a single token representing the latent embedding from HTS-AT? I suggest including a more detailed explanation of this aspect, rather than solely focusing on the right side of Figure 3.**\
> We thank the reviewer for the question. The length of `prefix_1` and `prefix_2` is 40 each. The output from HTS-AT is a single embedding (dim=768), which is then converted into `[seq_len, d]` using audio projection. The audio projection (right side of Fig 3) first expands the hidden dimension to a larger size dimension `k`, which is then split to form `[seq_len, d]` where `k = seq_len * d`. This is followed by concatenating with a learnable constant, resulting in `[seq_len + c, d]`. This output is then passed to the transformer, followed by clipping of the learnable constant output `c`. The resulting output of audio projection is of shape `[seq_len, d]`. This architecture of audio projection, specifically expanding a single embedding to seq_len, is inspired by previous Audio-Language Model papers published in NeurIPS 2023 and ICASSP 2024. We added this explanation in Section 3.1 under "Audio Projection" in the main paper. We hope this clarifies the reviewer's question.

---

> > ### Author Response · Authors · 2024-11-23
> > **Request to review the rebuttal**
> >
> > Thank you reviewer uxzd for reviewing our paper. We have addressed your questions in our rebuttal response and uploaded a revised version of the paper. As the rebuttal period is nearing its conclusion, we kindly request you to review our rebuttal and share any additional comments or concerns you may have. Thank you once again for your valuable feedback!

---

> > > ### Comment · Reviewer_uxzd · 2024-11-25
> > >
> > > Thank the authors for the response.
> > > I am satisfied with the response and this solves my questions. I raises the score to 8.

---

### Official Review · Reviewer_EdMr · 2024-11-05

**Soundness:** 2
**Presentation:** 2
**Contribution:** 3
**Rating:** 8
**Confidence:** 4

**Summary:**

This paper proposes a new task  - Audio Difference Explanation, which involves generating natural language description of differences between two audio recordings. They motivate this new task by pointing its significance in fields like audio forensics, audio quality assessment, and synthetic audio generation. They point out that  the current landscape of Audio-Language models like SALMONN and GAMA are geared towards applications like audio question answering, audio captioning, etc but havent addressed audio difference explanation.

The authors introduce new datasets Audio Caps Difference (ACD) and Clotho Difference (CLD). These contain pair of audios with text representing difference between the audios generated by an LLM. The text descriptions are at three levels of detail: concise, brief, detailed. These descriptions are human verified and hallucination corrected for the test subset.

The authors propose a model ADIFF to perform this task. This model encodes the two audios, followed by a projection layer to project these embeddings in LLM space. inserts a separator token in the latent space and concatenates the BPE tokenized representations from a question of the form "Explain the difference between the two audios". They pass this through a cross-projection layer, the output of which serves as an prompt to the LLM backbone (GPT2 in their work), which in turn produces a textual description of the difference between the audios. The authors indicate a 3 stage training of their model - 1. unimodal training of audio encoder, LLM (using pretrained models suffice), 2. audio grounding in language and 3. finetuning all components.

The authors provide ablations on how the cross projection layer is important. As a baseline, authors propose a "naive" baseline - which is similar to existing work in literature. The authors also perform both objective and subjective evaluations of their proposed model, and perform various ablations to understand it better.

**Strengths:**

- Introduction of a new task - The paper addresses a previously underexplored area by defining the audio exploration task and the paper motivates this task well.

- Dataset creation and availability -  The authors took great care in creating new datasets ACD and CLD for this task - which is a significant contribution for motivating further research on this task.

- Novelty in Model Architecture - the authors build upon existing literature to introduce a new cross projection layer that helps compare the two audios properly - a component which is absent in existing literature on audios.

- Comprehensive Evaluation - The authors do both subjective and objective evaluations to establish the models performance. The authors also include an ablation with random weights for Audio encoder - thus establishing a lower bound on the objective evaluations.

- Hallucination detection tools - Authors provide means to detect hallucinations in the models responses using audio event predictions via a HTSAT encoder.

**Weaknesses:**

- Absence of a concrete baseline - the baseline the authors compare ADIFF against is a clearly inferior version of the same model, and thus guaranteeing that ADIFF would perform better than this baseline. This baseline is more of an ablation of the components of the model.


- A major contribution of this work is "cross projection" layer that distinguishes ADIFF from baseline and existing literature. However there is Insufficient evidence of importance of the cross projection layer due to two factors:

1. The authors motivate that cross projection layer is very important by utilizing the bottom two sections of Table 4 where middle section represents  baseline model with frozen LLM, and lower section indicate baseline model + cross projection layer and sep token. But in the main model ADIFF, they finetune the LLM too, which could also learn what cross-projection learns, hence making having a cross projection layer redundant. If the comparison was between  baseline + LLM finetuned vs baseline + cross projection layer + LLM finetuned, it would indicate more conclusively that cross posting is useful.

2. Authors use GPT2 as the LLM model which ranges from a few hundred million params to 1.5 billion params. Its unclear which LLM was used for the cross projection ablation. Arguably, a larger LLM + finetuning could learn the task of cross projection layer. Its been shown that prompt tuning is very powerful for models like Llama3 8B models, and this ablation might not yield any difference there.


- The paper’s structure and presentation can be improved. There is redundancy and lack of detail in tables and captions —such as the overlapping results in Tables 2 and 4, missing key information about the LLM model used, and insufficient explanations of scores presented in Table 3, which collectively make it challenging to interpret the results accurately.

- The paper inaccurately equates compute across different-sized models in section 5.3 by training them for the same number of epochs, failing to account for the higher per-epoch computational cost of larger models. This undermines the  validity of their scaling comparisons in this section.

**Questions:**

- Its unclear from the paper why  Qwen-Audio Chat is a valid baseline for subjective evaluation in Table 3. The authors mention that its the only ALM that supports two audio inputs, but thats not a reason enough to use this as a baseline for subjective evals if it is not designed to do audio difference explanation task?

- Insufficient information on human evaluation - its unclear how many evaluations were done here, how many reviewers were involved and how it was ensured that the evaluation is fair. Its unclear what the numbers in Table 3 mean, and hence deriving insights out of them is hard.

- Authors mention "ADIFF sees the largest improvement in granularity metric", but this is not very clear from the table 3 itself, a more detailed table, maybe in appendix that highlights this could be more readable.

- Its unclear why the datasets used for Table 3 different from ACD and CLD datasets?

- Its unclear how the LLM is finetuned. Is a full finetune performed? How about using PEFT  instead and utilizing larger models?

---

> ### Author Response · Authors · 2024-11-14
> **Rebuttal by Authors (1/2)**
>
> We sincerely thank the reviewer for the detailed comments. We carefully address each of the reviewer’s concerns below and hope that our response resolves your concerns. Any follow-up questions are welcome.
>
> ---
> **Absence of a concrete baseline - the baseline the ... components of the model.**\
> We appreciate the reviewer's suggestion and have addressed it in the global comment question 1. We use the existing literature on prefix-tuning as the baseline for the audio difference explanation benchmark and systematically improve upon it, showing consistent performance improvements. Finally, the ADIFF model significantly outperforms the naive baseline. Nevertheless, we understand the reviewer's point and have added two more baselines in the objective and subjective evaluation section: First, Qwen-AC with LoRA fine-tuned on the training set of ACD and CLD, and second, Qwen-AC with full fine-tuning (no frozen parameters) on the training set of ACD and CLD. The results are shown in Table 2 and Table 3, with analysis in Section 4.2. We hope this addresses the concerns on the baseline.
>
> **The authors motivate that cross projection layer ... that cross posting is useful.**\
> The reviewers bring up a valid point. We would like to address it by dividing the question into two parts. (1) *How useful is the cross-projection layer?* The utility of cross-projection is highlighted in Table 4, where the middle section represents the baseline model with a frozen LLM, and the last section shows the baseline model with the addition of the cross-projection layer and sep token. These results indicate that cross-projection is a useful addition in prefix-tuning architecture where the LLM is frozen. We also performed a qualitative study that shows how this helps the model utilize text prefixes to store difference information, as detailed in Appendix I. (2) *Now given that the LM is also fine-tuned, does the cross-projection layer retain its utility?* This is a valid question, and we thank the reviewer for bringing this up. To verify this, we conducted an ablation study suggested by the reviewer, comparing the proposed final ADIFF model (three-stage training with cross-projection) against ADIFF (three-stage training but without cross-projection). The results and analysis are added in Appendix N. We see that the performance improvement from cross-projection is reduced when the LM is also fine-tuned, but ADIFF with cross-projection still outperforms ADIFF without it on all three tiers of ACD and CLD. We hypothesize that the cross-projection acts as block attention over the two audio and text prefixes, which has been shown to improve performance in the vision domain [1]. We hope this clarifies the reviewer's question.
>
> **Authors use GPT2 as the LLM model which ranges ... ablation might not yield any difference there.**\
> We apologize for any confusion. We use the GPT2-base (128M) in all experiments, including cross-projection. Only in the scaling section (Section 5.3) do we vary the LM size from 128M to 1.5B. While we do not disagree that a larger LLM with fine-tuning might learn the task of cross-projection, we did not scale to 8B pairs and thus cannot conclusively comment on this. We did our experiments from the perspective of building small audio-language models in the future, as audio tasks are generally performed on-device (embedded) where running large LLMs is not feasible. Nonetheless, we reinforce that the contribution of cross-projection to prefix tuning and even fine-tuning remains valid from the experiments conducted in the paper and the new ablation study added in Appendix N.
>
> **The paper’s structure and presentation can be improved ... to interpret the results accurately.**\
> We apologize for any confusion and space limitations. We have addressed this in the global comment question 2. We combined the results in Tables 2 and 4 for ease of comparison. The Table 3 scores and their meaning were moved to Appendix H2 for lack of space. We will ensure that pointers to the appendix are explicitly specified in the main text.
>
> **The paper inaccurately equates compute across ... their scaling comparisons in this section.**\
> We thank the reviewer for bringing this up. We explicitly state in Section 5.3 - "Each model is trained with the same compute budget, approximately equivalent to 30 epochs". The only conclusion we draw in Section 5.3 is the last two statements "This suggests that aligning and guiding larger models with prefix-tuning requires a greater number of epochs. Given the computation budget and performance, we choose to use GPT2-base for the subsequent experiments." We agree with the reviewer that higher per-epoch computational costs result in increased compute requirements for larger LMs. We have updated the Section 5.3 and added this discussion. Our conclusion (in quotes) for the section still holds and becomes even stronger as small LMs require even less compute.
>
> [1] https://arxiv.org/abs/2407.07726

---

> ### Author Response · Authors · 2024-11-14
> **Rebuttal by Authors (2/2)**
>
> **Its unclear from the paper why Qwen-Audio Chat is a valid ... do audio difference explanation task?**\
> The reviewer brings up a valid point and we have addressed this in global comment question 1. We trained Qwen-Audio Chat on the ACD and CLD datasets to create a stronger baseline. We used both LoRA fine-tuning and full fine-tuning and added the results to Table 2 and Table 3. We hope this addresses the concern regarding Qwen-Audio Chat not being designed for the task of audio difference explanation.
>
> **Authors mention "ADIFF sees the largest improvement in ... could be more readable.**\
> The Table 3 scores and their meaning are described in Appendix H2. We hope this clarifies the numbers, and we will add better pointers to Appendix Section H2.
>
> **Its unclear why the datasets used for Table 3 different from ACD and CLD datasets?**\
> The reviewer raises a valid question. We specifically chose to source audio files from different datasets instead of ACD and CLD for human evaluation for two reasons. First, we wanted to compare against other Audio-Language Models like Qwen-Audio, which supports two audios. Training the naive-baseline and ADIFF on the training set of ACD and CLD would inherently match the distribution of the test dataset in terms of both audio and language used, which is unfair to Qwen-Audio, which might not have seen similar audio recordings. Therefore, to ensure a fair comparison, we excluded audios from the ACD and CLD test set in the human evaluation. Second, we aimed to observe how performance changes across different types of audios, such as different acoustic scenes, same audio events, different domains, etc. By sourcing audios from Studio, FSD50K, and GTZAN, we were able to achieve the desired level of control and simulate the scenarios explained in Section 4. We hope this helps!
>
> **Its unclear how the LLM is finetuned. Is a full ... larger models?**\
> The LM has been fully fine-tuned. We did not explore using PEFT on larger LLMs. We hope that the Qwen-Audio-Chat with PEFT (LoRA) fine-tuning provided above covers this baseline. If not, we are willing to conduct additional experiments that are within are compute budget.

---

> > ### Author Response · Authors · 2024-11-23
> > **Request to review the rebuttal**
> >
> > Thank you reviewer EdMr for reviewing our paper. We have addressed your questions in our rebuttal response and uploaded a revised version of the paper. As the rebuttal period is nearing its conclusion, we kindly request you to review our rebuttal and share any additional comments or concerns you may have. Thank you once again for your valuable feedback!

---

> > > ### Author Response · Authors · 2024-11-26
> > > **Request to review the rebuttal**
> > >
> > > Thank you reviewer EdMr for reviewing our paper. We have addressed your questions in our rebuttal response and uploaded a revised version of the paper. As the rebuttal period ends in a day, we kindly request you to review our rebuttal and share any additional comments or concerns you may have. Thank you once again for your valuable feedback!

---

> > > > ### Comment · Reviewer_EdMr · 2024-11-29
> > > >
> > > > Thank you for addressing all my questions and concerns, additional experiments and improvement to the manuscript structure. Thank you for doing this in a timely manner too, I appreciate the authors for their diligence and hard work, I am convinced about the utility of this work and am raising my  score to 8.

---

### Author Response · Authors · 2024-11-14
**Author Rebuttal**

We sincerely thank all the reviewer for recognizing our contribution and providing constructive feedback, especially for acknowledging that "**The paper introduces a considerable novelty by defining a new audio task**" (Reviewer uxzd), "**The proposed task in audio difference is novel and interesting and good contribution to the audio reasoning research community.**" (Reviewer 6ejz), "**Dataset creation and availability ... which is a significant contribution for motivating further research on this task**" (Reviewer EdMr), "**The authors do both subjective and objective evaluations to establish the models performance.**" (Reviewer EdMr), and **thorough ablation studies in proposed architectures**" (Reviewer 6ejz).

---
We would like to re-emphasize the novelty and technical contributions of this work. We present the task of explaining audio differences and subsequently propose benchmarks and baselines for the task. Our key contributions are: (1) New Audio Task and Datasets: we introduce a novel audio difference explanation task, motivating its relevance and need within the field. Creation of new datasets (ACD and CLD) with three tiers which are verified by human annotators for the test set. (2) Introduce ADIFF, which extends the naive prefix tuning baseline with a novel cross projection layer, three-step finetuning process, leading to consistent improvement across all metrics. (3) Thorough evaluation including subjective and objective methods, ablation studies, tools for detecting hallucinations, and presenting our findings for motivating subsequent works in small audio-language models.

We summarize the two main questions brought up by the reviewers and address them here: \
**Question 1: Stronger baseline to compare against ADIFF. Finetuning the SOTA model Qwen-Audiofor this task using PEFT and full-finetuning.**\
We appreciate the reviewer's suggestion and assistance in improving our work. We selected Qwen-AC as the SoTA baseline because it is the only ALM in the literature that supports two audio inputs. However, we agree that it makes sense to fine-tune Qwen-AC on the proposed dataset and include it as a baseline. We have conducted the suggested experiments, fine-tuning Qwen-AC on the proposed data using Parameter-Efficient Fine-Tuning (LoRA) and full fine-tuning. The details have been added to Section 4.1, with the test set results of the proposed dataset included in Table 2 and the results with subjective evaluation presented in Table 3. Additionally, we expanded Section 4.2 to provide an analysis of these results and compare them against ADIFF. We hope these additional experiments address the concern about stronger baseline.

**Question 2: The paper’s result presentation can be improved. Showing the complete captioning metrics in all the tables (2, 4, 5, 6) does not seem to help in the narratives, especially not many of them are referred in the text. It might worth consolidate and only show a couple. Also it might be helpful to bold the high performant numbers that are also referred to in the text.**\
We appreciate the reviewer’s suggestion. We have updated the paper, specifically the result tables (Table 2, 4, 5, 6), to improve readability. First, we retained only the BLUE_4, METEOR, and SPIDEr metrics in the main paper's result tables as suggested by reviewer 6ejz. We did not include SPICE and CIDEr in the main result table as its linear combination is SPIDEr. The complete tables including SPICE and CIDEr metrics are available in the appendix and are referenced in the main paper's result tables. Second, in the new format, we bolded the best-performing numbers for each metric across models on the ACD and CLD tasks for each Tier. We hope these changes significantly improve the readability and ease of interpreting the results.

---

### Meta-Review · Area_Chair_92nT · 2024-12-20

**Metareview:**

The paper deals with explaining audio differences between pairs of recordings. It presents a new audio difference explanation task, trained using two datasets. The explanation was designed according to three tiers and verified by human annotators. The audio difference explanation system ADIFF  takes two audio recordings and text prompt as input, encoding and projecting the audio embeddings to a latent space through a cross-projection layer, which is then used as a prompt to a transformer decoder to generate natural language explanation. The experimental section includes subjective and objective evaluation, ablation studies, and other special tasks. Issues of hallucinations are mentioned as concern to be addressed.

**Additional Comments On Reviewer Discussion:**

All reviewers seemed to acknowledge the importance and originality of the task and validity of the solution. Specific question about the baseline, and technical details of the ADIFF system and other concerns were addressed by the authors to reviewers satisfaction. Notably, the baseline experiments were augmented during rebuttal by adding Qwen-AC with LoRA fine-tuned, and Qwen-AC with full fine-tuning on the training set.

---

### Decision · Program_Chairs · 2025-01-22

Accept (Spotlight)